# Water release and homogenization by dynamic recrystallization of quartz

Junichi Fukuda[1], Takamoto Okudaira[1], Yukiko Ohtomo[2]

[1]Department of Geosciences, Osaka Metropolitan University, Osaka, 558-8585, Japan
[2]Department of Education, Art and Science, Yamagata University, Yamagata, 990-8560, Japan

*Correspondence to*: Junichi Fukuda (jfukuda@omu.ac.jp)

**Abstract.** To evaluate changes in water distribution generated by dynamic recrystallization of quartz, we performed infrared (IR) spectroscopy mapping of quartz in deformed granite from the Wariyama uplift zone in NE Japan. We analyzed three granite samples with different degrees of
deformation: almost undeformed, weakly deformed, and strongly deformed. Dynamically recrystallized quartz grains with a grain size of ~10 µm are found in these three samples, but the percentages of recrystallized grains and recrystallization processes are different. Quartz in the almost undeformed sample shows wavy grain boundaries with a few bulged quartz grains. In the weakly deformed sample, bulging of quartz, which consumed adjacent host quartz grains, forms regions of
a few hundred micrometers. In the strongly deformed sample, almost all quartz grains are recrystallized by subgrain rotation. IR spectra of quartz in the three samples commonly show a broad water band owing to $H_2O$ fluid at 2800–3750 $cm^{-1}$ with no structural OH bands. Water contents in host quartz grains in the almost undeformed sample are in the range of 40–1750 wt. ppm, with a mean of $500 \pm 280$ wt. ppm $H_2O$. On the other hand, water contents in regions of recrystallized
grains, regardless of the recrystallization processes involved, are in the range of 100–510 wt. ppm, with a mean of $220 \pm 70$ wt. ppm, which are low and homogeneous compared with contents in host quartz grains. These low water contents in recrystallized regions also contrast with those of up to 1540 wt. ppm in adjacent host grains in the weakly deformed sample. Water contents in regions of subgrains are intermediate between those in host and recrystallized grains. These results for water
distribution in quartz imply that water was released by dynamic recrystallization.

**1 Introduction**

Quartz is a major constituent of Earth's crust, and its deformation can control crustal rheology (e.g., Rutter and Brodie, 1988; Kohlstedt et al., 1995; Bürgmann and Dresen, 2008; Fossen and Cavalcante, 2017). For plastic deformation of quartz, water occurring as an impurity in this mineral can reduce its strength by up to several orders of magnitude, as has been confirmed by numerous experimental investigations (e.g., Griggs and Blacic, 1965; Griggs, 1967; Blacic, 1975; Parrish et al., 1976; Jaoul et al., 1984; Kronenberg and Tullis, 1984; Koch et al., 1989; Post et al., 1996; Chernak et al., 2009; Holyoke and Kronenberg, 2013). In addition, increasing water fugacity reduces the plastic strength of quartz (Post et al., 1996; Chernak et al., 2009; Holyoke and Kronenberg, 2013; Fukuda and Shimizu, 2017; Fukuda et al., 2018; Tokle et al., 2019; Lu and Jiang, 2019; Lusk et al., 2021; Nègre et al., 2021). As a species of water, molecular $H_2O$ is trapped as fluid inclusions and at grain boundaries, and the crystal structure of quartz incorporates OH groups that are coupled with impurity cations in quartz (Kats, 1962; Aines and Rossman, 1984; Kronenberg, 1994; Stenina, 2004; Stalder, 2021; also see the review by Stünitz et al., 2017, for water species in quartz and their effects on plastic deformation).

Water in naturally deformed quartz has been measured by infrared (IR) spectroscopy focusing on contents and species ($H_2O$ or OH) (Kronenberg and Wolf, 1990; Kronenberg et al., 1990; Kronenberg, 1994; Nakashima et al., 1995; Niimi et al., 1999; Muto et al., 2004, 2005; O'Kane et al., 2007; Gleason and DeSisto, 2008; Menegon et al., 2011; Finch et al., 2016; Kilian et al., 2016; Kronenberg et al., 2017, 2020; Fukuda and Shimizu, 2019). However, the effects of water content and species in quartz on the strength of quartz are not fully understood, especially regarding the quantitative relationships between water content, species, and flow laws of quartz. These relationships in olivine have been partly evaluated by experimental means (Jung and Karato, 2001; Hirth and Kohlstedt, 2003; Katayama et al., 2004; Jung et al., 2006; Masuti et al., 2019) and have been linked to the deformation of natural olivine and to mantle dynamics (Karato et al., 2008; Katayama et al., 2011; Skemer et al., 2013; Jung, 2017).

IR spectroscopic measurements of water in quartz aggregates have revealed that water content increases with decreasing grain size (Kronenberg and Wolf, 1990; Kronenberg et al., 1990; Kronenberg, 1994; Nakashima et al., 1995; Muto et al., 2004, 2005; Fukuda and Shimizu, 2019), which has been explained in terms of volume increases of grain boundary regions being able to host more water than grain interiors (Ito and Nakashima, 2002). It is also known that grain size can be

reduced by dynamic recrystallization (reviewed for quartz in, e.g., Stipp et al., 2002; Tullis, 2002; Passchier and Trouw, 2005). Recent studies of naturally deformed quartz by Finch et al. (2016) and Kronenberg et al. (2020) have shown that water content decreases as grain size decreases by dynamic recrystallization, on which basis those authors proposed that water is released by dynamic recrystallization. The release of water by dynamic recrystallization has also been shown by the experiments of Palazzin et al. (2018). Those various studies performed point IR analyses in regions composed of dynamically recrystallized quartz grains. However, there is still only limited information on water contents, species, and distributions with the development of microstructure, meaning that more data are required for water distributions in naturally deformed quartz.

In this study, we performed IR mapping analyses of undeformed quartz grains and dynamically recrystallized grains. We use the results as a basis for discussion of variations in water content and the behavior of water release arising from microstructural change due to dynamic recrystallization.

## 2 Samples

Granite samples with differing degrees of deformation were obtained from the Wariyama uplift zone in NE Japan, which is located on the eastern side of the Futaba Fault (Fig. 1). The geological setting of this zone has been reported by Oide and Fujita (1975) and Fujita et al. (1988). The Wariyama uplift zone shows a N–S to NNE–SSW trend and includes many vertical faults (Fujita et al., 1988). Tsuchiya et al. (2014) reported a U–Pb zircon age of ca. 300 Ma for the Wariyama granite and documented an age of ca. 120 Ma for the Takase granite to the eastern side of the Wariyama granite. In the uplift zone, the deformed mylonitic Wariyama granite is found on the eastern side of the Takase pass as a local shear zone (Tsuchiya et al., 2013). The western and eastern edges of the Wariyama uplift zone are bounded by the NNW–SSE-trending Futaba Fault Zone, which is a sinistral strike-slip fault that formed during the mid-Cretaceous (Fujita et al., 1988; Otsuki, 1992). The foliations of the mylonitic rocks strike N–S to NNE–SSW with the dip to the east and the stretching lineations gently plunge to the north or south. Thus, the deformation of the Wariyama granite may be caused by the mid-Cretaceous movement of the Futaba Fault Zone. We divided the deformed Wariyama granite samples into three types—almost undeformed, weakly deformed, and strongly deformed—on the basis of the development of foliations of mylonitic rocks as observed in hand specimen (Fig. 2) and of microstructures under a polarizing microscope (Figs 3 and 4).

In the almost undeformed and weakly deformed samples, mafic igneous minerals are partly replaced by epidote and chlorite (Fig. 3a–d). Although some or all chlorite may have replaced amphibole during late-stage hydrothermal alteration in the three types of sample (Fig. 3a, c, and e), chlorite is found in shear bands in the strongly deformed sample (Fig. 3e), suggesting that chlorite may have been stable during mylonitization. Grain interiors of plagioclase are commonly altered to epidote, muscovite, and clay minerals in the three types of sample. In the weakly and strongly deformed samples (Fig. 3c–f), brittle deformation of plagioclase dominates and plagioclase and amphibole form porphyroclasts, but there is no development of microstructures indicative of dynamic recrystallization or pressure shadows at their margins. In the strongly deformed samples, amphibole (mainly hornblende) and epidote grains may have been stable during mylonitization because shear bands developed in the samples are composed of amphibole and epidote grains (Fig. 3e and f). According to Fujita et al. (1988), some metasedimentary rocks affected by the thermal effects of granitoid intrusions in the Wariyama area contain metamorphic andalusite, implying that these granitoids were emplaced into upper-crustal levels. On the basis of these observations, mylonitization at least of the strongly deformed sample is inferred to have occurred under epidote–amphibolite-facies conditions and within or near the andalusite stability field (i.e., ~500 °C; Spear, 1993).

In the microstructures of quartz, original quartz grains of up to 2 mm in size are observed (Fig. 4a). However, quartz grain boundaries are wavy, and a few bulged grains of ~10 µm occur (Fig. 4b). Therefore, we term this sample "almost" undeformed. In original quartz grains, fluid inclusions are observed and are heterogeneously distributed (Fig. 4b). In the weakly deformed sample cut parallel to the lineation and perpendicular to the foliation, larger regions of a few hundred micrometers are composed of bulged quartz grains that have developed from adjacent elongated host quartz grains (Fig. 4c and d). In comparison with the almost undeformed sample, there are fewer large fluid inclusions. Thus, the redistribution of fluid inclusions by deformation is inferred, as reported previously in natural samples (Kerrich, 1976; Wilkins and Barkas, 1978; Hollister, 1990; Cordier et al., 1994; Vityk et al., 2000; Faleiros et al., 2010). According to those previous studies, fluid inclusions may undergo leakage or smaller inclusions or structural OH may be formed, as confirmed experimentally by Stünitz et al. (2017) and Palazzin et al. (2018). In this study, we examine the actual water distribution by using IR spectroscopic measurements. The strongly deformed sample shows well-developed foliations (Fig. 2) and almost fully recrystallized quartz microstructure (Fig.

4e and f). Recrystallized quartz grains measuring ~10 µm in size and formed by subgrain rotation constitute the major part of the quartz matrix (Fig. 4e). In some regions of quartz, crystallographic orientations continuously change as in undulose extinction, and these regions are composed of

subgrains (Fig. 4f). Differences in the recrystallization mechanisms of quartz in the three types of sample may be due to differences in temperature, strain rate, and/or stress.

Grain sizes of recrystallized grains were determined by image analysis of traced grains. More than 300 grains of recrystallized grains in optical photomicrographs (e.g., Fig. 4b, d, and f) were traced. The area of each grain was determined by image analysis using ImageJ software, and grain

size was expressed as the diameter of a circle of equivalent area to the diameter of the grain. The mean grain sizes of recrystallized quartz with ±1 standard deviation of the almost undeformed, weakly deformed, and strongly deformed samples are $10.5 \pm 3.7$, $6.7 \pm 1.9$, and $9.6 \pm 3.6$ µm, respectively (Fig. 5). The piezometers of Stipp and Tullis (2003) and Holyoke and Kronenberg (2010) developed mainly for bulging recrystallization were applied to the mean grain sizes of the

almost undeformed and weakly deformed samples and they yield values of a few tens of megapascals. Similarly, values of a few hundred megapascals are obtained for the mean grain size of the strongly deformed sample using the piezometers of these two papers for subgrain rotation (and for grain boundary migration, but this type of recrystallization is not evident in our samples). Values of a few hundred megapascals are also obtained from all of the samples for a grain size of

~10 µm, for which the reported piezometers do not distinguish the mechanisms of dynamic recrystallization (Twiss, 1977; Shimizu, 2008, 2012; Cross et al., 2017).

## 3 Analytical procedure for IR spectroscopy

Quartz grains in the almost undeformed, weakly deformed, and strongly deformed samples were measured by IR mapping to evaluate water contents and species. Sample thin sections with an

approximate thickness of 100 µm were prepared: The process for making thin sections followed that for regular thin sections with a thickness of ~30 µm for observation using a polarizing microscope but there are a few exceptions: One side of the sample was polished down to #6000 aluminum oxide powder and glycol phthalate resin was used to attach the polished sample surface on a glass slide on a hot plate heated to 100–120 °C. Then, the other side of the sample section was polished down to

#6000 to give an approximate thickness of 100 µm to minimize scattered IR light on the sample surface and interference fringes within the sample. Finally, the sample was removed from the glass

slide by dissolving the resin in acetone. IR spectra obtained in this study did not show any CH peaks around 2950 cm$^{-1}$, which can be indicative of organic contamination from residuals of the resin and/or acetone; even if these peaks are detected in IR spectra, they do not affect water bands, as the wavenumbers are different (e.g., Kebukawa et al., 2009). Subsequently, the sample thin section was set on a stand stage equipped with a dial gauge and the actual sample thicknesses of the IR mapped areas were measured. The sample thickness of approximately 100 µm is suitable for obtaining good transmitted IR signals through quartz and for making microstructural observations under a polarizing microscope. Water in tight grain boundaries and triple junctions (e.g., at the scale of transmission electron microscope resolution), as well as water in grain interiors, is preserved during the above-described processes of thin sectioning and heating on a hot plate (Fukuda et al., 2009). Therefore, arguments and interpretations in this paper concern water that was originally trapped in the samples.

We used a Fourier-transform IR spectrometer (JASCO FT/IR-4700) equipped with a microspectrometer (JASCO IRT-5200) to measure water included in quartz and other minerals in the samples. The IR spectrometer incorporated a silicon carbide (Globar) IR source and KBr beamsplitter. Unpolarized IR light was irradiated to the sample set on the sample stage of the microspectrometer, and IR light through the sample under atmospheric conditions was detected using a mercury–cadmium–telluride detector in the microspectrometer. IR spectra were obtained from an average of 100 scans with a wavenumber resolution of 4 cm$^{-1}$. An aperture size of $25 \times 25$ µm was used. Mapping measurements were performed using a beam-moving function in the sample area of up to $400 \times 400$ µm; the sample stage was fixed, and IR light irradiated to the sample was moved. Mapping step intervals the same as the aperture size were applied. We checked that mapping measurements with larger aperture sizes for the same areas gave similar averaged water distributions. We also included other minerals such as plagioclase and/or phyllosilicate neighboring quartz in the mapping, as their IR spectra differ substantially from those of quartz and can serve as markers in IR mapped data to match with the locations in the optical photomicrographs.

Water contents were converted from absorbance of water stretching bands with linear baselines from 2800 to 3750 cm$^{-1}$. Water stretching bands in our quartz samples commonly exhibit molecular $H_2O$ bands (presented below in the Result section). We used the calibration of Paterson (1982) designed for molecular $H_2O$ as $C = \int A(\nu)/100(3780 - \nu)d\nu$, where $C$ is the concentration in mol $H_2O$ $\ell^{-1}$ (Note that the original paper of Paterson (1982) uses H $\ell^{-1}$), and $A(\nu)$ is the absorbance at a wavenumber $\nu$ in cm$^{-1}$ for a sample thickness normalized to 1 cm. This calibration is based on a

linear trend of absorption coefficients for any type of water ($H_2O$ and OH) in different materials but assumes isotropically distributed molecular $H_2O$ for the orientation factor of 1/3, which is already included in the above equation. The mapped areas also include plagioclase and/or phyllosilicate. Grain interiors of plagioclase are also altered to epidote, muscovite, and clay minerals (Fig. 3 and see the description in Section 2). IR spectra of plagioclase possibly including these alteration minerals as well as those of phyllosilicate exhibit a dominant band owing to molecular $H_2O$ and accessory band(s) caused by structural OH (shown later). Especially, structural OH may be anisotropically incorporated in all of these minerals. It is noted, therefore, that the calibration of Paterson (1982) used for these minerals gives semi-quantitative water contents. Other calibrations for contents of molecular $H_2O$ and/or OH species in quartz have been given by Kats (1962), Aines et al. (1984), Nakashima et al. (1995), Libowitzky and Rossman (1997), Stipp et al. (2006), and Thomas et al. (2009). Fukuda and Shimizu (2019) compared these calibrations, and the ratios of calculated water contents between them based on Paterson (1982) as unity are 0.68:0.88:1.15:1.31:1.58:0.43. Previously reported absorption coefficients of water in quartz have been discussed by Fukuda and Shimizu (2019) and Stalder (2021). In this study, water contents are expressed as wt. ppm $H_2O$, which is converted from mol $H_2O$ $\ell^{-1}$ using the molar mass of $H_2O$ (18 g $mol^{-1}$) and the density of quartz (2650 g $\ell^{-1}$). This conversion using these values was applied to other minerals, namely, plagioclase and/or phyllosilicate, which are commonly measured together with quartz in IR maps and used as markers for the locations measured. As the densities (averaged densities when mixed) of these minerals differ from that of quartz, albeit slightly, the water contents of the minerals reported in this study are semi-quantitative values.

## 4 Results

The results of IR mapping for the almost undeformed sample are shown in Fig. 6. The measured area includes a single quartz grain (Fig. 6a). Water contents are heterogeneous (Fig. 6b), ranging from a minimum of 40 wt. ppm (No. 1) to a maximum of 1750 wt. ppm (No. 5), the raw IR spectra for which are presented in Fig. 6c. The heterogeneous distribution of fluid inclusions is clearly observed in the optical photomicrograph (Fig. 4b) and would correspond to the IR mapped data. Raw IR spectra commonly show a broad band at 2800–3750 $cm^{-1}$, which is a result of the stretching vibrations of molecular $H_2O$ (Fig. 6c).

The results of IR mapping for the weakly deformed sample are presented in Fig. 7. The measured area includes two host quartz grains, dynamically recrystallized grains around them, and plagioclase grains (Fig. 7a). In the water distribution map (Fig. 7b), plagioclase shows much higher water contents (around No. 5) compared with quartz. The higher water contents in plagioclase in the mapped data serve as a marker for comparison with the microstructural image (Fig. 7a). The IR spectrum of plagioclase in Fig. 7c shows a sharp band at 3630 cm$^{-1}$, which may be related to structural hydroxyl vibration(s) of plagioclase and/or altered clay minerals (Johnson and Rossman, 2004). In all of the studied samples, grain interiors of plagioclase are altered to epidote, muscovite, and clay minerals (Fig. 3). When ductile shear zones developed in the study area, plagioclase was not deformed plastically but was replaced by various minerals during late-stage alteration. Therefore, the IR spectra of plagioclase that includes alteration minerals may not provide useful information about water in rocks during mylonitization. Water contents in the host quartz grains are in the range of 600–1000 wt. ppm (Fig. 7b). In contrast, water contents in the recrystallized regions of >~150 µm distance from the adjacent host grains are 200–300 wt. ppm, clearly lower than those in the host grains. In addition, a gradual decrease in water content from the host grains to the adjacent recrystallized regions is observed in some cases; for example, over a distance of ~150 µm from No. 3 (host; 680 wt. ppm in Fig. 7c) to No. 1 (recrystallized region; 210 wt. ppm). In terms of microstructure, this decrease is consistent with the development of dynamic recrystallization from subgrains in and around host grains (Fig. 7a) to recrystallized grains.

In another part of the weakly deformed sample (Fig. 8), regions of small grains around host grains widen at certain sample stage angles under a polarizing microscope and subgrains would be included (Fig. 8a–f). Water contents in recrystallized regions probably including subgrains are 290–500 wt. ppm (around Nos 1 and 2 in Fig. 8g and h), slightly higher than those in the recrystallized regions in Fig. 7. These regions look slightly darker under plane-polarizing light, probably because of light scattering by fine grains (Fig. 8c and e). We did not observe microcracks that hold water, at least under a polarizing microscope. Water contents in host grains range up to 1540 wt. ppm, which is as high as those in the almost undeformed sample.

The results of IR mapping for the strongly deformed sample are shown in Fig. 9. The IR mapped area consists mainly of recrystallized quartz (Fig. 9a). Crystallographic orientations of some grains within and around a relic of a host grain look continuously similar and/or continuously change, as in undulose extinction, and these grains would be subgrains (inset under cross-polarized

light in Fig. 9a). Water contents in the subgrain region (around No. 4 in Fig. 9b) are 390–540 wt. ppm, substantially higher than those in other recrystallized regions (water contents of 120–250 wt.

240 ppm). The IR spectra of quartz in the strongly deformed sample commonly show a broad $H_2O$ band at 2800–3750 cm$^{-1}$ (Fig. 9c), the same as quartz in the almost undeformed and weakly deformed samples (Figs 6–8). The low and homogeneous water contents in the recrystallized regions are consistent with those in the weakly deformed sample (Fig. 7).

## 5 Discussion

### 5.1 Water in host quartz and water species

Water contents in host quartz grains are heterogeneous regardless of the degree of deformation (Figs 6–8) and range from 40 to 1750 wt. ppm with a mean of $500 \pm 280$ wt. ppm in the almost undeformed sample and from 250 to 1540 wt. ppm with a mean of $800 \pm 200$ wt. ppm in the weakly deformed sample (Fig. 10). The IR spectra commonly show a broad band at 2800–3750 cm$^{-1}$ as molecular

$H_2O$ in the three different samples. In the almost undeformed sample, fluid inclusions in quartz grains are visible and heterogeneously distributed in the optical photomicrograph (Fig. 4b), but submicroscopic fluid inclusions may be included. In the weakly deformed sample, fluid inclusions in host quartz grains are less numerous in the photomicrograph (Fig. 4d). In the IR mapping images for the weakly deformed sample, the high-water-content regions (Figs 7b and 8g) correspond to

relics of host grains identified under a polarizing microscope (Figs 7a and 8a–f). In comparison, in the almost undeformed sample, water contents in host grains are simply heterogeneous. As water contents in host quartz grains in the almost undeformed and weakly deformed samples do not differ substantially, although their distributions in the two samples are different, a redistribution of fluid inclusions with a size change to less than the optical microscopic scale must have occurred in the

latter sample. The redistribution of fluid inclusions owing to plastic deformation has been reported in previous microstructural observations (Kerrich, 1976; Wilkins and Barkas, 1978; Hollister, 1990; Cordier et al., 1994; Vityk et al., 2000; Faleiros et al., 2010): The redistribution of fluid inclusions is possible by volume diffusion and/or diffusion through dislocation cores (pipe diffusion) of water molecules and/or hydrogen. Some of those studies proposed leakage of water during the

redistribution of fluid inclusions, but this is unlikely to have been significant in the host grains analyzed in the present study since the water contents in host grains in the almost undeformed and weakly deformed samples are not significantly different. The redistribution of fluid inclusions has

similarly also been reported from IR spectroscopic measurements of an experimentally deformed single quartz crystal (Stünitz et al., 2017). Those authors demonstrated that original fluid inclusions with sizes of up to 100 µm transformed into Si–OH as Si–O–Si + $H_2O$ ← → Si–OH $\cdots$ OH–Si, where Si–OH showed a sharp band at 3585 cm$^{-1}$ in IR spectra. This band was also observed in naturally deformed quartz by Niimi et al. (1999) and Gleason and DeSisto (2008) but was not detected in our samples or others (e.g., Kronenberg and Wolf, 1990; Kronenberg et al., 1990; Muto et al., 2004, 2005; O'Kane et al., 2007; Finch et al., 2016; Kronenberg et al., 2017, 2020; Fukuda and Shimizu, 2019). Stünitz et al. (2017) also demonstrated that the above Si–OH, which could have originally been visible fluid inclusions with sizes of a few micrometers under a polarizing microscope, are again transformed into much smaller fluid inclusions measuring less than 100 nm in size by subsequent annealing. This process may have occurred in our samples and can explain the redistribution of fluid inclusions in the optical photomicrographs (Fig. 4b vs. 4d) while retaining the original water contents.

## 5.2 Water in dynamically recrystallized quartz

As seen in the weakly and strongly deformed samples, where dynamic recrystallization of quartz is developed, water contents in recrystallized regions are low and homogeneous compared with those in host grains (Figs 7–9). The mean water content is 220 ± 70 wt. ppm (Fig. 10) where grains are fully recrystallized. These low water contents also contrast with those of up to 1540 wt. ppm in adjacent host grains in the weakly deformed sample. As the sizes of the recrystallized grains are ~10 µm (Fig. 5), and the sample thickness is ~100 µm, the aperture size of 25 × 25 µm of the IR microspectrometer captures water in both grain interiors and grain boundaries. In the regions of subgrains in the weakly deformed sample (Figs 7 and 8) and in the strongly deformed sample (Fig. 9), water contents in subgrain regions are intermediate between those in recrystallized regions and host grains. Thus, water in quartz is inferred to be released by dynamic recrystallization. Similar results have been reported by Finch et al. (2016), Kilian et al. (2016), and Kronenberg et al. (2020) for naturally recrystallized quartz and by Palazzin et al. (2018) for experimentally recrystallized quartz. Kronenberg et al. (2020) reported that water contents in quartzites from the Moine thrust vary from around 600 to 220 wt. ppm depending on the percentage of recrystallized regions from 20% to 100% toward the thrust within a distance of 70 m. The mean recrystallized grain size in their samples was 21.7 µm. As those authors performed point analyses, it may be difficult to correlate the

water content values with the microstructures, and their water content values may thus include subgrains, which are more likely to be measured when recrystallization percentages are low. In addition, in our weakly deformed sample, water contents in quartz gradually decrease with increasing distance from the adjacent host grains within a range of ~150 µm (Fig. 7), which may be due to the transition from subgrains to recrystallized grains. In addition, in the strongly deformed sample, water contents in regions of subgrains are higher than those in recrystallized regions (Fig. 9). Water contents reported by Finch et al. (2016) for deformed granitic diatexite vary from approximately 650 to 200 wt. ppm from their weakly deformed sample to mylonite to ultramylonite and with decreasing grain size from 95 to 60 µm (there are also slight differences in water contents between ribbon and matrix regions in their samples). Palazzin et al. (2018) reported water contents in matrix regions with a grain size of <10 µm in experimentally deformed quartzite of ~200 wt. ppm, even though different starting quartz samples with different water contents ranging from 39 to 3446 wt. ppm were used (their table 1). Kilian et al. (2016) studied quartz in a granite deformed under amphibolite-facies conditions. In their samples, quartz grains were dynamically recrystallized to a grain size of 250–750 µm mainly by grain boundary migration with a minor contribution by subgrain rotation. Their mean water contents, including intracrystalline OH and molecular $H_2O$ as fluid inclusions in recrystallized quartz, were ~10 wt. ppm, comprising 70%–80% of molecular $H_2O$ and the remainder of intracrystalline OH, and were approximately half of those in the original magmatic host grains. Those values in recrystallized and original grains are much lower than those in other previous studies and in the present study. Thus, water contents in recrystallized quartz grains and host grains are diverse in natural samples; this diversity may depend on deformation conditions, including the equilibrium state of water. In addition, Fukuda et al. (2012) found heterogeneously distributed water with contents of 150–2200 wt. ppm in K-feldspar porphyroclasts in granitoid mylonite. In contrast, in the same granitoid mylonite sample, water in regions of fine-grained K-felspar (~20 µm), which could have been formed by solution–precipitation processes around porphyroclasts, was homogeneously distributed with contents of 150–300 wt. ppm, much lower than those in porphyroclasts. Those authors also proposed that water was released during the formation process of fine grains.

If the water content of $220 \pm 70$ wt. ppm in the recrystallized regions analyzed in this study is distributed homogeneously in grain boundaries as thin films, then the mean grain size of ~10 µm (Fig. 5) gives a grain boundary width of ~2 nm based on the estimation by Ito and Nakashima (2002)

(1.9 nm using their cubic grain model and 2.5 nm using their tetradecahedral grain model). This is consistent with the estimation by Palazzin et al. (2018) for their samples whose grain sizes are similar to ours. However, there is no basis for assuming homogeneous grain boundary films. Some previous studies of natural quartz aggregates have shown that water contents increase with decreasing grain size (Kronenberg et al., 1990; Nakashima et al., 1995; Ito and Nakashima, 2002; Muto et al., 2004, 2005; Fukuda and Shimizu, 2019). In these cases, where quartz displays microstructural modifications by dynamic recrystallization and/or grain growth to change grain size, water may have been continuously supplied to fill grain boundaries, where more water can be stored compared with grain interiors. Thus, IR spectra of these samples measured in laboratories may show large water absorption bands stored in grain boundaries with decreasing grain size compared with samples that did not undergo subsequent water infiltration along grain boundaries in previous studies (Finch et al., 2016; Kilian et al., 2016; Kronenberg et al., 2020). Under natural conditions, including those of the upper and middle crust, fluid water could form thin films in quartz grain boundaries with a wetting angle of <60°, as confirmed by experiments (Watson and Brenan, 1987; Holness, 1992, 1993) and observations of natural samples (Hiraga et al., 1999). For example, in the IR mapping measurements reported by Fukuda and Shimizu (2019) for quartz in Sanbagawa metamorphic schist, the estimated water content varies from 310 to 40 wt. ppm in quartz aggregated regions with changing grain size from ~40 µm (chlorite zone) to ~120 µm (oligoclase–biotite zone). The water content in the latter sample is inferred to be from grain interiors, as the grain size and sample thickness were sufficiently large for the aperture sizes of $30 \times 30$ µm or $50 \times 50$ µm used in those authors' study. For their samples whose grain sizes were smaller than the aperture sizes, the measured water would include grain boundaries. Their estimated grain boundary width filled with water was ~10 nm, which is approximately five times larger than that estimated in the present study. However, grain boundaries are not always filled with water, depending on the water-available conditions after dynamic recrystallization. In our samples, quartz grain boundaries may not have undergone a subsequent water infiltration process, and the water distribution indicates that water contents in recrystallized regions include the water released from host grains by dynamic recrystallization.

### 5.3 Mechanism of water release and homogenization by dynamic recrystallization

In our samples, it is clear that water occurring as fluid inclusions in host quartz grains is released and homogenized by dynamic recrystallization (Figs 6–10). Fluid inclusions in host quartz grains may be trapped during quartz crystallization from magma. As experimentally confirmed by Tarantola et al. (2010), when plastic strain is very low (<1%), the shapes of fluid inclusions may be elongated or burst in the original locations. In contrast, under higher plastic strain when dislocations, which would be associated with the formation of hydroxyl, are introduced as $Si–O–Si + H_2O \longleftrightarrow$ $Si–OH \cdots OH–Si$, water from original fluid inclusions can be redistributed by diffusion, which is also rate-limited by this reaction (Kerrich, 1976; Wilkins and Barkas, 1978; Hollister, 1990; Cordier et al., 1994; Vityk et al., 2000; Faleiros et al., 2010). In addition to this type of diffusion, diffusion through dislocation cores (i.e., pipe diffusion) can redistribute fluid inclusions (Trepied and Doukhan, 1978; Cordier et al., 1988, 1994; Kronenberg, 1994). In cases where Si–OH is preserved, the IR spectrum shows its OH vibration at 3585 cm$^{-1}$ (Stünitz et al., 2017) or at 3596 cm$^{-1}$ (Niimi et al., 1999; Gleason and DeSisto, 2008). However, as discussed above, Si–OH transforms back to $SiO_2 + H_2O$ by annealing to form smaller fluid inclusions with sizes of <100 nm both in host and recrystallized grains (Stünitz et al., 2017). Thus, the redistribution of fluid inclusions caused by diffusion within host grains may account for water distributions that correspond to the shapes of host grains in the weakly deformed sample (Figs 7 and 8), as discussed in section 5.1. In cases where diffused $H_2O$ during the redistribution process reaches newly formed grain boundaries by dynamic recrystallization, this water should be released, as grain boundaries are fast diffusion paths that lead to the equilibrium state of water in the system. In other words, original fluid inclusions in host quartz grains in the photomicrograph (Fig. 4b) and IR spectra (Fig. 6) were trapped during quartz crystallization from magma, and their contents were not equilibrated with the water system under deformation conditions (possibly including water activity and/or water fugacity). During the dynamic recrystallization process from subgrains to fully recrystallized grains, the water release process may be enhanced by the development of fast diffusion paths. In fact, water contents in regions of subgrains are intermediate between those in host grains and regions of recrystallized grains (Figs 7–9), indicating that the degree of water release depends on the degree of dynamic recrystallization. Some studies have also speculated that water transfer including release can occur through microcracking (e.g., Kronenberg et al., 1986; Palazzin et al., 2018), but this process was not confirmed in our samples. In summary, through dynamic recrystallization, water release and

homogenization occur to achieve the equilibrium state of water in the system corresponding to the deformation conditions.

**6 Conclusions**

We performed IR mapping measurements on host and recrystallized quartz in three types of Wariyama granite: almost undeformed, weakly deformed, and strongly deformed. Our results allow the following conclusions to be drawn:

• Water in host quartz grains in the almost undeformed sample has contents of 40–1750 wt. ppm, with a mean of $500 \pm 280$ wt. ppm, and is heterogeneously distributed as fluid inclusions.

• Fluid inclusions in host quartz grains in the weakly deformed sample become less numerous under a polarizing microscope compared with those in almost undeformed sample. The water contents of host quartz grains in the two samples do not change substantially, but the water distributions in host grains in the weakly deformed sample can correspond to the shapes of host grains. This suggests a redistribution of water within host grains during plastic deformation.

• When dynamic recrystallization occurs, water is mostly likely released along newly formed grain boundaries by diffusion and homogenized to achieve an equilibrium state. Water contents in regions of recrystallized grains are 100–510 wt. ppm, with a mean of $220 \pm 70$ wt. ppm, in the weakly and strongly deformed samples, whereas those of adjacent host grains observed in the weakly deformed sample are up to 1540 wt. ppm.

• Water contents in regions of subgrains are intermediate between those in host grains and recrystallized regions in the weakly and strongly deformed samples, implying that the degree of water release depends on the degree of dynamic recrystallization.

*Data availability.* Raw IR mapped data for Figs 6–9 in JASCO format (e.g., Fig6.jwa) and each exported IR spectrum for each sample position (e.g., Fig6_X1Y1.txt) are available from Mendeley Data, V1, doi: 10.17632/zn24kbg9xt.1 *at https://data.mendeley.com/datasets/zn24kbg9xt/draft?a=3552fbc3-364c-4edd-b50a-fa58d198bcce [Comment from authors: This is a temporary shared URL. We will officially publish the data in Mendeley Data when the manuscript is published as an article in Solid Earth].* The text files can be

used for those who do not have JASCO's IR software. These data can be used to reproduce the water content variations presented in Fig. 10.

*Sample availability.* The samples used in this study are available at the locations shown in Fig. 1. The rock specimens for Fig. 2 and thin sections for Figs 3 and 4, and Figs 6–9 (IR analyses) are stored in the laboratory of JF. The thin sections for Fig. 4 were used for the grain size analyses

presented in Fig. 5.

*Author contributions.* JF conceptualized the study, performed the formal analyses for all data, and carries overall responsibility for the study. JF and YO investigated the geological setting. All authors investigated the sample microstructures. JF and TO analyzed the IR data. JF drafted the main part of the paper. All authors drafted, reviewed, and edited the paper.

*Competing interests.* The authors declare that they have no conflicts of interest.

*Acknowledgments.* The authors thank the constructive reviews and encouragement given by two anonymous referees and A. Kronenberg. We also thank T. Hirono for his support of our IR measurements. This research was supported by a Grant-in-Aid for Scientific Research (KAKENHI 19K04041 for JF and 20K04087 for TO) provided by the Japan Society for the Promotion of Science

(JSPS).

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

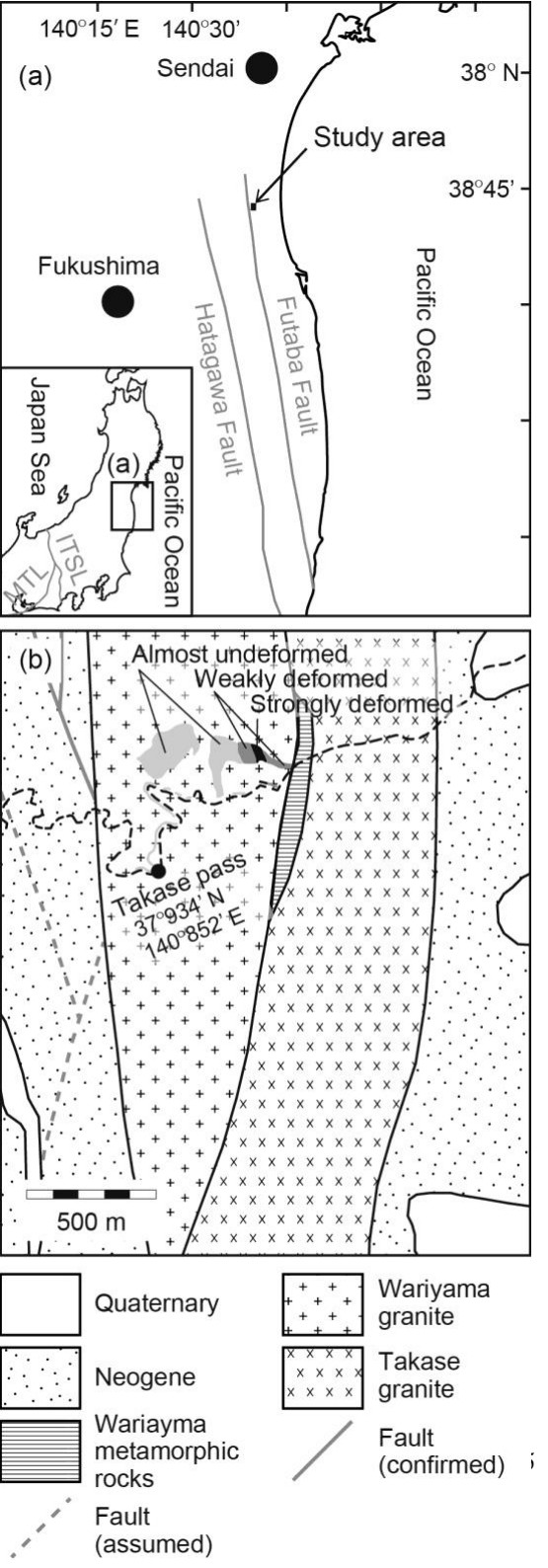

## (a)

140°15′ E  140°30′

Sendai

Study area

38° N

38°45′

Fukushima

Pacific Ocean

Hatagawa Fault

Futaba Fault

Japan Sea

(a)

Pacific Ocean

ITSL

MTL

## (b)

Almost undeformed
Weakly deformed
Strongly deformed

Takase pass
37°934′ N
140°852′ E

500 m

Quaternary

Neogene

Wariayma
metamorphic
rocks

Fault
(assumed)

Wariyama
granite

Takase
granite

Fault
(confirmed)

**Figure 1. Study area location map (a) and geological map (b) of northeastern Japan after Oide and Fujita (1975), Fujita et al. (1988), and Tsuchiya et al. (2014). The inset in (a) shows the location of (a) in Japan and major tectonic lines [the Itoigawa–Shizuoka Tectonic Line (ISTL) and the Median Tectonic Line (MTL)]. In (b), the dashed black line represents a path including Takase pass. Solid black lines represent geological boundaries. The Wariyama granite was categorized into three types based on the degree of deformation, as observed in hand specimen (Fig. 2) and by microstructures observed under a polarizing microscope (Figs 3 and4). The three types are referred to as "almost undeformed", "weakly deformed", and "strongly deformed", with distributions as shown in the geological map.**

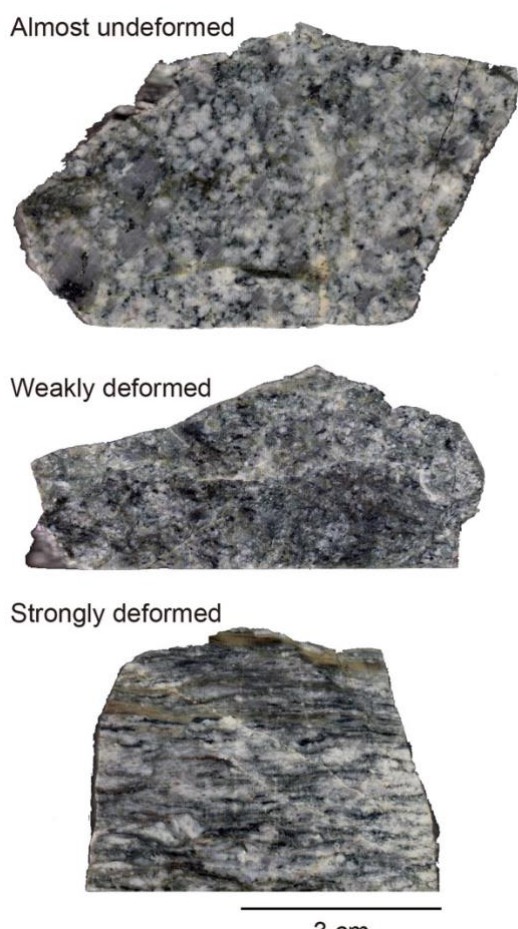

**Figure 2. Scanned images of the three studied samples of the Wariyama granite. The two deformed samples were cut parallel to the lineation and perpendicular to the foliation.**

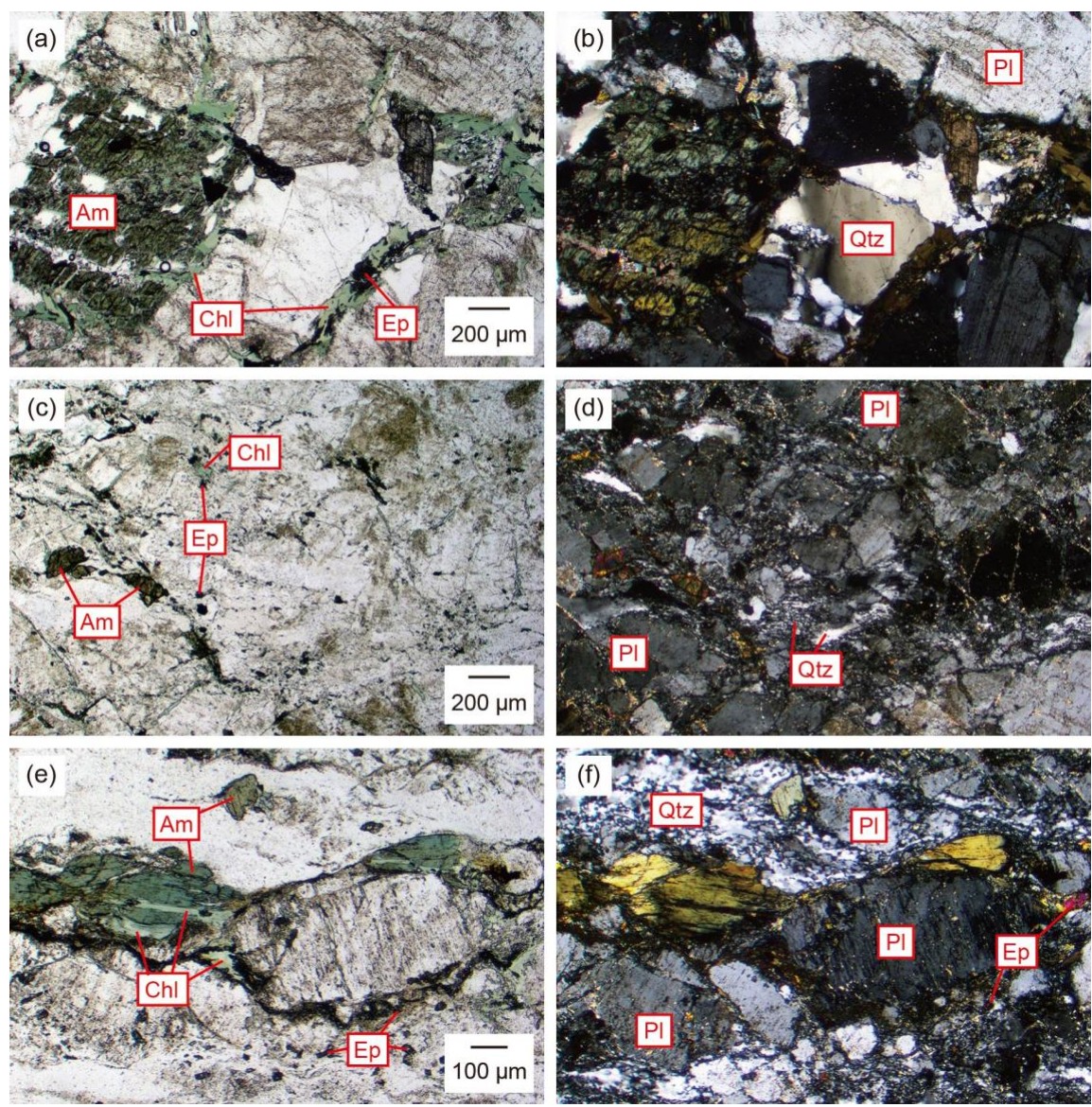

Figure 3. Optical photomicrographs showing microstructures of the Wariyama granite under plane-polarized (a, c, and e) and cross-polarized (b, d, and f) light. (a and b) Almost undeformed sample. (c and d) Weakly deformed sample. (e and f) Strongly deformed sample. Microstructures of quartz in the three samples are shown in Fig. 4.

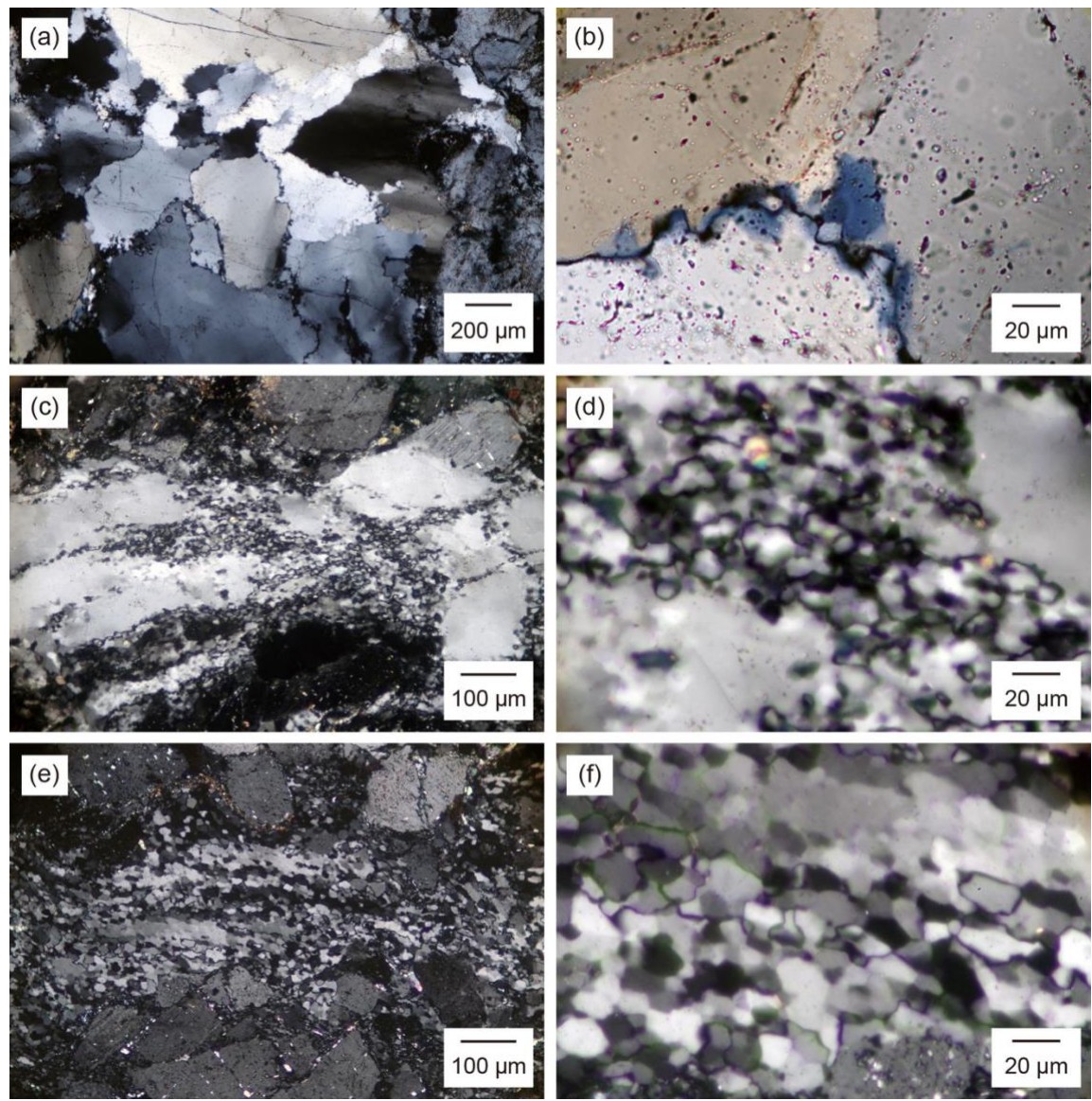

Figure 4. Optical photomicrographs taken under cross-polarized light showing quartz microstructures in the Wariyama granite. (a and b) Almost undeformed sample. Quartz grain boundaries are wavy, and a few bulged grains are observed. (c and d) Weakly deformed sample cut parallel to the lineation and perpendicular to the foliation. Bulging of quartz, which consumed adjacent host quartz grains, forms regions of a few hundred micrometers, and host grains are elongated. (e and f) Strongly deformed sample cut parallel to the lineation and perpendicular to the foliation. Almost all quartz grains are recrystallized by subgrain rotation, and subgrains remain in some regions, as in the upper part of (f).

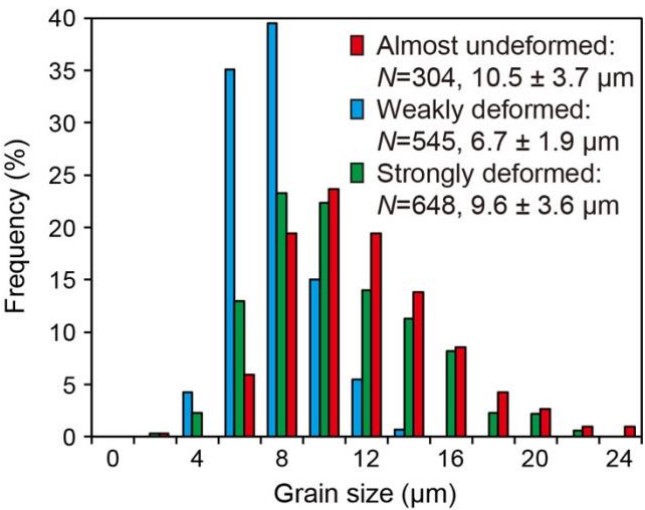

**Figure 5. Grain size distributions of recrystallized quartz grains of the studied samples with three different degrees of deformation.** $N$ **is the number of grains measured. Mean grain sizes and standard deviations are given.**

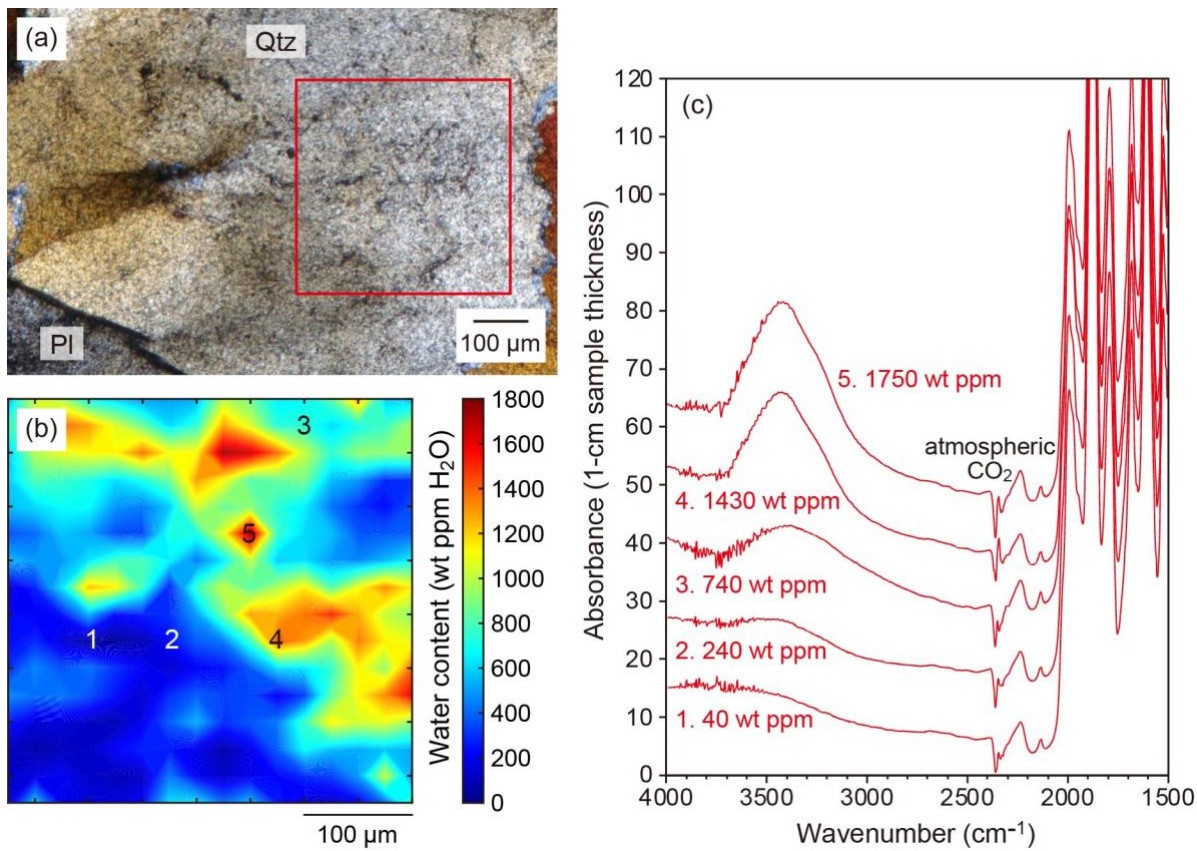

**Figure 6. IR mapping results for the almost undeformed sample. The sample thickness is 60 μm. (a)**
715 **Optical photomicrograph under cross-polarized light. The IR mapped area is shown with a red square.**
**(b) Water distribution. Numbers represent raw IR spectra in (c) from low to high water contents.**

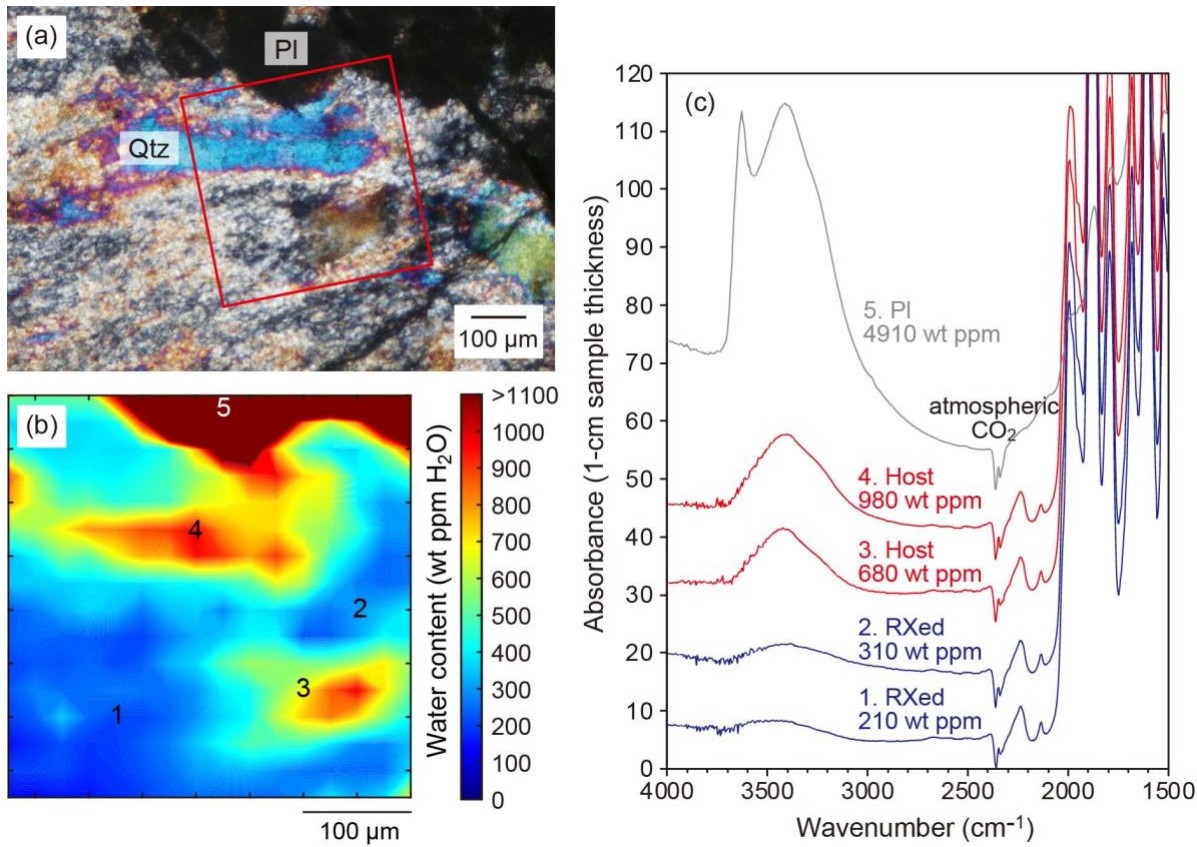

**Figure 7. IR mapping results for the weakly deformed sample. The sample thickness is 103 μm. (a) Optical photomicrograph under cross-polarized light. The IR mapped area is shown with a red square. (b) Water distribution. Numbers represent raw IR spectra in (c). Color-coded water contents up to 1100 wt. ppm $H_2O$ are for quartz, and those above are for plagioclase around No. 5 in (b). "RXed" in (c) means "recrystallized".**

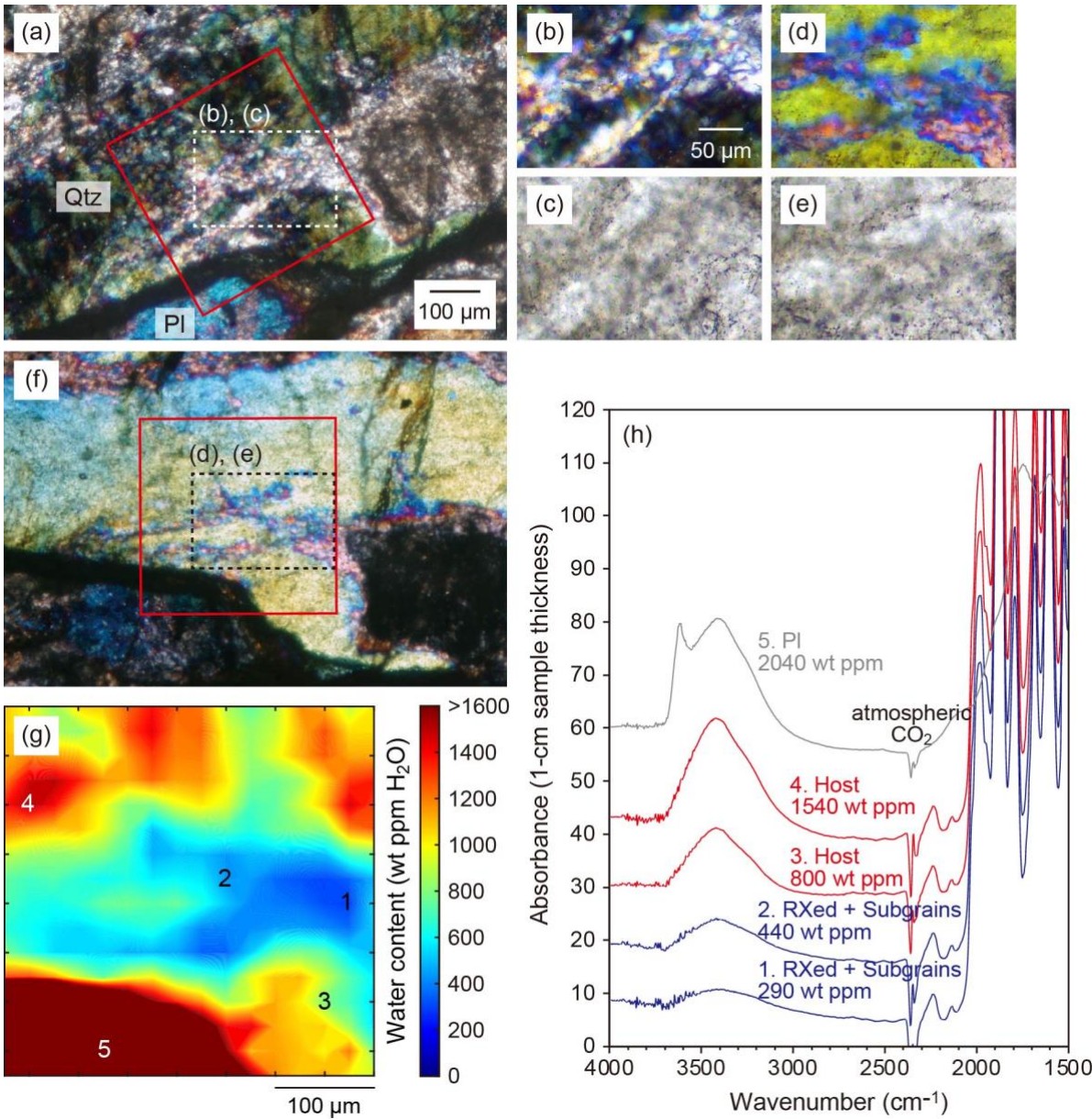

**Figure 8. IR mapping results for the weakly deformed sample. The sample thickness is 97 μm. (a) Optical photomicrograph under cross-polarized light. The IR mapped area is shown with a red square. The dashed white rectangle depicts the area of enlarged images of fine-grained quartz regions including host grains under (b) cross-polarized and (c) plane-polarized light. Similarly, images (d) and (e) are from (f), where the optical photomicrograph is rotated approximately 30° clockwise from (a). Fine-grained quartz regions are smaller than those in (a), indicating subgrains are included. (g) Water distribution. Numbers represent raw IR spectra in (h). Color-coded water contents up to 1600 wt. ppm $H_2O$ are for quartz, and**

those above are for plagioclase around No. 5 in (g). "RXed" in (h) means "recrystallized". Both recrystallized grains and subgrains are included in the measured area.

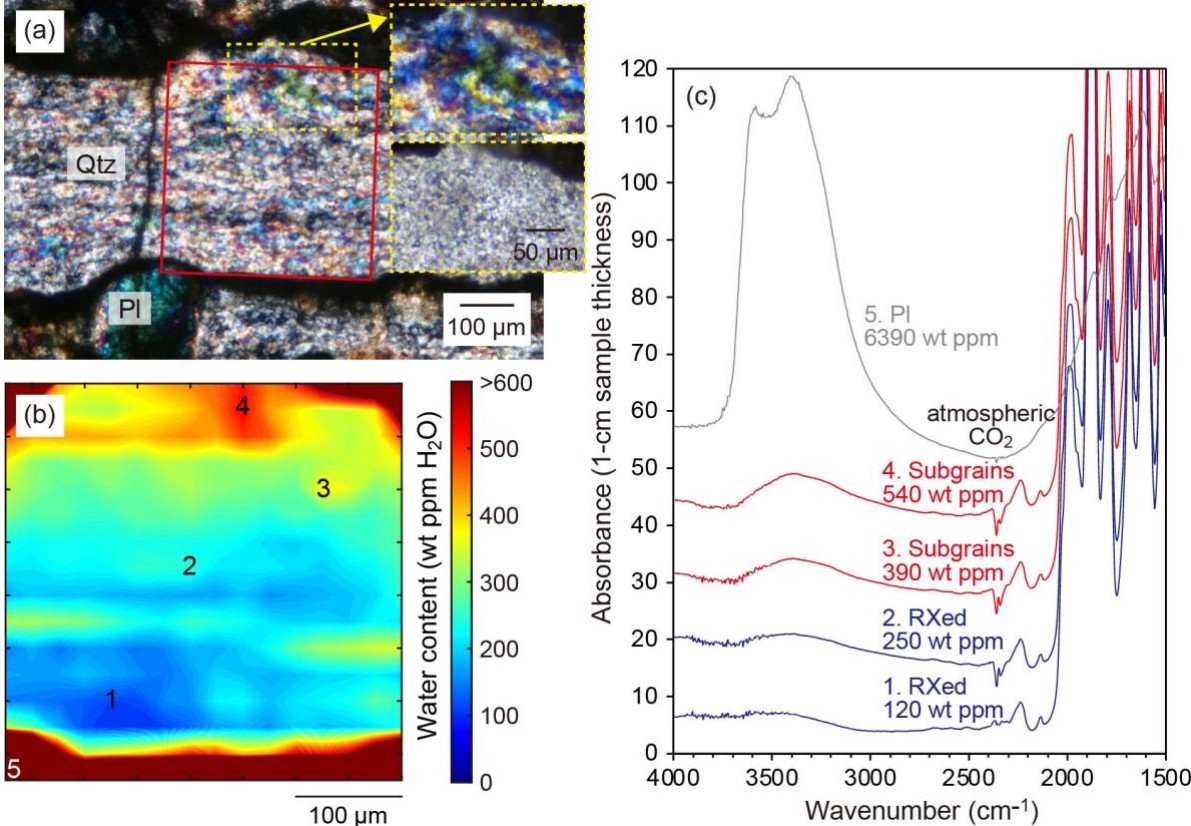

Figure 9. IR mapping results for the strongly deformed sample. The sample thickness is 112 μm. (a) Optical photomicrograph under cross-polarized light. The IR mapped area is shown with a red square. The two insets in the yellow rectangles show enlarged images of a region that includes a relic of a host grain at the center under cross-polarized (upper) and plane-polarized (lower) light. (b) Water distribution. Numbers represent raw IR spectra in (c). Color-coded water contents up to 600 wt. ppm
$H_2O$ are for quartz, and those above are for plagioclase and phyllosilicate around No. 5 and others in (b). Quartz subgrains in host grains are around Nos 3 and 4 in (b), which correspond to the microstructure in (a). "RXed" in (c) means "recrystallized".

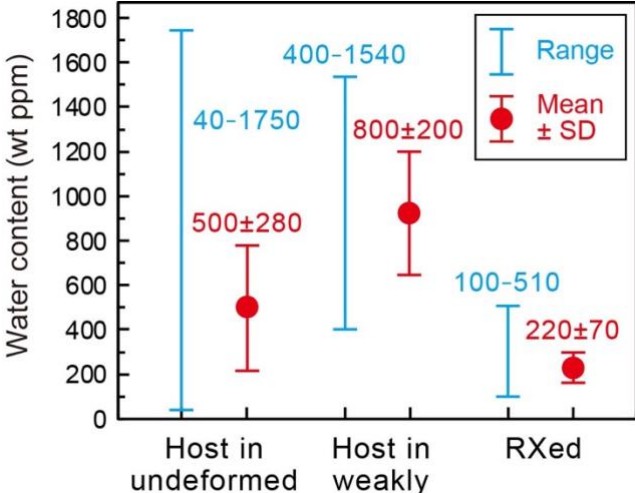

**Figure 10. Summary of water content variations in quartz. "Host" represents host quartz grains as grain interiors in the almost undeformed and weakly deformed samples. These two samples are referred to as "undeformed" and "weakly" in the figure, respectively. "RXed" represents regions of recrystallized quartz grains including both grain interiors and grain boundaries in the weakly deformed and strongly deformed samples. Ranges from the lowest to highest contents in blue and means with standard deviations (SD) in red are shown. The difference in water contents in host grains between the almost undeformed and weakly deformed samples may be because of the areas to be measured. In addition, the water distributions in host grains in the weakly deformed sample correspond to the shapes of host grains (Figs 7 and 8), but those in the almost undeformed sample do not (Fig. 6). Therefore, there may be a redistribution of water in host grains by plastic deformation, as supported by the photomicrographs, which show less visible fluid inclusions in the latter sample (Fig. 4).**