# Peer review of "Water release and homogenization by dynamic recrystallization of quartz"

_EGUsphere, 2022_

## Author Response (AR1)

Referee's comments: Black (Arial)

Our replies: Blue (Arial)

Revisions made to the manuscript in response to the referees' comments: Red (Times New Roman).

Descriptions from the original manuscript: Black (Times New Roman)

**Replies to Referee #1's comments**

These comments are intended for the editor and authors.

In this short paper, Fukuda et al. compare the water content of quartz in three granitic samples using infrared spectroscopy. Water has a major affect on the rheology of quartz, but the details of this are complicated and not well understood. The paper is fairly clearly written, the figures are of excellent quality, and I enjoyed reading the paper and learned some things. I must point out here that while I am familiar with recrystallization of quartz, I have not studied water contents in quartz and the reader of this review should keep that in mind while considering my comments.

My main concern about the paper is the small sample size (N=3) and use of only one technique to investigate the samples. As far as I know, IR measurements are not particularly expensive or laborious to make, although IR mapping may be relatively novel (I'm not sure). So my sense is that the study is well below the average published contribution in terms of the intellectual rigor involved in its production. Also, other studies have already shown similar results—the authors cite two previous studies (Finch et al., 2016; and Kronenberg et al., 2020) showing decreased water concentration associated with recrystallization in natural samples. Adding another data set like this to the literature is valuable, however I am accustomed in published work to see significantly more data presented and/or a more detailed analysis that makes more progress towards some outstanding question. There does not seem to any specific question being targeted or addressed by the study, other than "how does water content in this shear zone change during recrystallization?".

IR point analyses of deformed quartz were performed in the 1990s during pioneering work by Prof. Kronenberg (Referee #3 of this manuscript). However, as IR spectroscopy is not commonly used and the IR mapping technique applied to water in deformed quartz is even rarer (Muto et al., 2004, 2005, Kronenberg et al., 2017; Fukuda and Shimizu, 2019), we still do not know much about water in natural quartz and its relationship to plastic deformation. With regard to these aspects, we need to know the following: 1. The water content of natural quartz; 2. whether water content changes as a result of textural development; and 3. whether water species change as a result of deformation. These questions should not be underestimated, as Prof. Kronenberg has commented in his review. Referee #1 states that our IR mapping measurements show only "water release by dynamic recrystallization", similar to the results by point analyses by Finch et al. (2016) and Kronenberg et al. (2020). However, the water

contents are different from those determined in previous studies, as discussed in section 5.2. In addition, our "mapping" measurements, which were not performed in those previous studies, show "water content distributions in host grains and in neighboring recrystallized regions". We also show the water-release process through intermediate water contents in regions of subgrains, which is also a new finding.

I recommend that additional samples are analyzed or some complimentary technique is added to the study before publication. For example, EBSD maps of the samples would allow a quantification of the degree of recrystallization and subgrain formation involved in changing water contents. Also, the authors infer the presence of subgrains in their thick sections, but this could be proven and quantified using EBSD. The authors also infer
different types of recrystallization in strongly deformed and weakly deformed samples which I find puzzling—such differences could also be quantified using EBSD.

With regard to "additional samples are analyzed", we originally performed IR mapping measurements on more than 20 areas of the three types of deformed rock, including different rock blocks and with the 100 μm aperture to test and strengthen our findings. In the manuscript, we report data that have unambiguous information (such as from other minerals and cracks). Some additional data are shown below, but we emphasize that water contents were calculated using the JASCO software and applying Paterson's (1982) simplified calibration (Fukuda and Shimizu 2019), which gave similar water contents to those calculated by applying Paterson's (1982) original calibration. Raw IR spectra show no difference from those presented in the manuscript, so they have been omitted. The data shown immediately below are similar to those in the manuscript and would be redundant even as supplementary data. Therefore, we prefer to keep them in this response file, as readers can also see this file.

Almost undeformed sample
Sample thickness: 95 µm

Weakly deformed sample
Sample thickness: 97 µm

[Figure]

Phyl: Phyllosilicate (biotite and/or chlorite)
Pl: Plagioclase

[Figure]

Strongly deformed sample
Sample thickness: 125 μm

Regarding EBSD measurements, we do agree with the referee's comment: EBSD measurements could prove the recrystallization process including the formation of subgrains as well as slip systems of quartz in the three types of our samples (almost undeformed, weakly, and strongly deformed samples). In our old papers, too (e.g., Fukuda et al. 2012 Tectonophysics; Fukuda et al. 2018; JGR; Fukuda et al. 2022 JSG), when IR spectroscopic measurements of water in rocks and minerals give supportive and/or additional information on deformation behavior, we showed other data by other measurements including detailed microstructural observations, EBSD measurements, and/or deformation experiments.

On the other hand, when IR spectroscopic measurements are the main topic of papers, we think we should focus only on them as other papers do (e.g., Kronenberg and Wolf 1990; Kronenberg et al. 1990; Kronenberg 1994; Nakashima et al. 1995; Niimi et al. 1999; Muto et al. 2004, 2005; Finch et al. 2016; Kronenberg et al. 2017, 2020; Fukuda and Shimizu 2019). This is also because other data by other measurements can be independent and other topics. Therefore, we suppose that the other two referees, who would be familiar with plastic deformation quartz and IR measurements, did not point out EBSD measurements.

**Below are some additional comments.**

13-15. Bulges form on host grains, so I don't understand how they can also be a few hundred microns distant. Please clarify the language.

We suppose that the referee pointed out that bulging is a local process so it can not be "a few hundred microns distant". We meant that bulged quartz grains in regions of a few hundred micrometers were a result of a repetitive bulging recrystallization as a local process consuming host grains. We have revised as follows.

L14,

bulging of quartz, which consumed adjacent host quartz grains, form regions of a few hundred micrometers.

We have also revised the similar expression in the Figure 3 caption.

L699,

Bulging of quartz, which consumed adjacent host quartz grains, form regions of a few hundred micrometers and host grains are elongated.

14-16. How can it be that small amounts of deformation involve bulging recrystallization, but large amounts of deformation are inferred to have experienced mainly subgrain rotation recrystallization? Is a switch in recrystallization mechanism over time being inferred, or did deformation occur at different conditions in different places (problematic for the study, since there is a tacit assumption that deformation conditions were similar in the three different samples). Alternatively, possibly there is not as much clarity about the deformation mechanisms as the author's think (EBSD analysis could help a little with this).

Rather than "a tacit assumption that deformation conditions were similar in the three different samples)", the deformation conditions (pressure, temperature, stress, and/or strain rate) may be different. In the original manuscript, we classified the recrystallization mechanisms based on the conventional microstructural observations under a polarizing microscope (e.g., Stipp et al. 2002 JSG; Tullis 2002 Rev in Mineral and Geochem).

In the revised manuscript, we have added a new figure as Fig. 3 which shows general microstructures of the three different samples and discussed their deformation conditions.

[Figure]

**Figure 3. Optical photomicrographs showing microstructures of the Wariyama granite under plane-polarized (a, c, and e) and cross-polarized (b, d, and f) light. (a and b) Almost undeformed sample. (c and d) Weakly deformed sample. (e and f) Strongly deformed sample. Microstructures of quartz in the three samples are shown in Fig. 4.**

L86,

In the almost undeformed and weakly deformed samples, mafic igneous minerals are partly replaced by epidote and chlorite (Fig. 3a–d). Although some or all chlorite may have replaced amphibole during late-stage hydrothermal alteration in the three types of sample (Fig. 3a, c, and e), chlorite is found in shear bands in the strongly deformed sample (Fig. 3e), suggesting that chlorite may have been stable during mylonitization. Grain interiors of plagioclase are commonly altered to epidote, muscovite,

and clay minerals in the three types of sample. In the weakly and strongly deformed samples (Fig. 3c–f), brittle deformation of plagioclase dominates and plagioclase and amphibole form porphyroclasts, but there is no development of microstructures indicative of dynamic recrystallization or pressure shadows at their margins. In the strongly deformed samples, amphibole (mainly hornblende) and epidote grains may have been stable during mylonitization because shear bands developed in the samples are composed of amphibole and epidote grains (Fig. 3e and f). According to Fujita et al. (1988), some metasedimentary rocks affected by the thermal effects of granitoid intrusions in the Wariyama area contain metamorphic andalusite, implying that these granitoids were emplaced into upper-crustal levels. On the basis of these observations, mylonitization at least of the strongly deformed sample is inferred to have occurred under epidote–amphibolite-facies conditions and within or near the andalusite stability field (i.e., ~500 °C; Spear, 1993).

L120,

Differences in the recrystallization mechanisms of quartz in the three types of sample may be due to differences in temperature, strain rate, and/or stress.

We have added Spear (1993) in the reference list.

20. Language issue: "is released" is problematic. "can be released" or "was released" would be better. There are scenarios where very dry quartz is recrystallized in the presence of water, and during this process water is added to the quartz. By saying "is" it sounds like the authors are making a universal claim.
We have changed "is released" to "was released" because this was what we found in our case and may not be "a universal claim" as the referee pointed out.

29 "several" better than "a few"
Revised.

Section 2 Samples: I would also like to see more attention paid to the deformation history of these samples. Are they from a strike slip, thrust, or normal sense deformation environment? Are they foot wall or hanging wall? Do we have information about the temperature of deformation? Is there information about the initial distance between these samples when deformation occurred (how similar do we really expect them to be)? Does previous work in the area suggest that the deformation of the three samples occurred simultaneously (at different strain rates), or was there a sequence of overprinting at progressively lower temperatures?
Regarding the deformation history and sense of deformation, we have clearly described them in the revised manuscript as follows.
L70,

Granite samples with differing degrees of deformation were obtained from the Wariyama uplift zone in NE Japan, which is located on the eastern side of the Futaba Fault (Fig. 1). The geological setting of this zone has been reported by Oide and Fujita (1975) and Fujita et al. (1988). The Wariyama uplift zone shows a N–S to NNE–SSW trend and includes many vertical faults (Fujita et al., 1988). Tsuchiya et al. (2014) reported a U–Pb zircon age of ca. 300 Ma for the Wariyama granite and documented an age of ca. 120 Ma for the Takase granite to the eastern side of the Wariyama granite. In the uplift zone, the deformed mylonitic Wariyama granite is found on the eastern side of the Takase pass as a local shear zone (Tsuchiya et al., 2013). The western and eastern edges of the Wariyama uplift zone are bounded by the NNW–SSE-trending Futaba Fault Zone, which is a sinistral strike-slip fault that formed during the mid-Cretaceous (Fujita et al., 1988; Otsuki, 1992). The foliations of the mylonitic rocks strike N–S to NNE–SSW with the dip to the east and the stretching lineations gently plunge to the north or south. Thus, the deformation of the Wariyama granite may be caused by the mid-Cretaceous movement of the Futaba Fault Zone.

We have added Otsuki (1992) in the reference list.

Regarding deformation conditions and their difference in the three different samples, we have added a new figure as Fig. 3 and discussed them. Please see our reply and revision regarding the referee's comment on L14-16.

96. "By imaging" is quite vague.
Indeed. We have revised as follows to summarize the sentences after that.
L122,
by image analysis of traced grains

101-102. These piezometers were calibrated using some samples that also experienced subgrain rotation recrystallization, not just bulging. Also, (line 106) the Cross piezometer uses a subset of the same samples Stipp used, so it's absurd to say that Cross et al is useful for samples that experienced subgrain rotation recrystallization, while Stipp piezometer is for bulging. In any case, it doesn't seem necessary to separate out two kinds of piezometers anyway because only rough estimates of stress are produced.
Even though "only rough estimates of stress are produced", "a few tens of megapascals" and "few hundred megapascals" seem quite different. Stipp and Tullis (2003) and Holyoke and Kronenberg (2010) determined a grain-size piezometer for regime 1 (based on mechanical data by Hirth and Tullis 1991, where the deformed quartz microstructures show mainly bulging) and another piezometer for regimes 2 and 3 together (regime 2 of subgrain rotation and regime 3 of grain boundary migration). Therefore, so long as the piezometers of those authors are used, their piezometers for the different recrystallization mechanisms need to be separated.

Given the referee's comment, we realized that we should explain the application of each piezometer to different recrystallization mechanisms. We have revised the relevant text as follows.

L128,

The piezometers of Stipp and Tullis (2003) and Holyoke and Kronenberg (2010) developed mainly for bulging recrystallization were applied to the mean grain sizes of the almost undeformed and weakly deformed samples and they yield values of a few tens of megapascals. Similarly, values of a few hundred megapascals are obtained for the mean grain size of the strongly deformed sample using the piezometers of these two papers for subgrain rotation (and for grain boundary migration, but this type of recrystallization is not evident in our samples). Values of a few hundred megapascals are also obtained from all of the samples for a grain size of ~10 μm, for which the reported piezometers do not distinguish the mechanisms of dynamic recrystallization (Twiss, 1977; Shimizu, 2008, 2012; Cross et al., 2017).

110-113. Inaccurate language: There is much more to preparing thin sections than by dissolving resin in acetone.

We have described more in the revised manuscript as follows.

L139,

Sample thin sections with an approximate thickness of 100 μm were prepared: The process for making thin sections followed that for regular thin sections with a thickness of ~30 μm for observation using a polarizing microscope but there are a few exceptions: One side of the sample was polished down to #6000 aluminum oxide powder and glycol phthalate resin was used to attach the polished sample surface on a glass slide on a hot plate heated to 100–120 °C. Then, the other side of the sample section was polished down to #6000 to give an approximate thickness of 100 μm to minimize scattered IR light on the sample surface and interference fringes within the sample. Finally, the sample was removed from the glass slide by dissolving the resin in acetone.

111-112. Similarly, this language about the dial gauge seems incomplete.

We have revised as follows.

L150,

Subsequently, the sample thin section was set on a stand stage equipped with a dial gauge and the actual sample thicknesses of the IR mapped areas were measured.

150-151. For those ignorant of such things (like myself): can we assume that the same IR calibration for quartz gives accurate values of water content in feldspars? A note and reference here or in the methods would be useful.

The calibration of Paterson (1982) we used is designed for molecular $H_2O$ but gives rough water contents in feldspar and phyllosilicate, which often measured together with quartz. We have clarified this as follows in the Methods section (3 Analytical procedure for IR spectroscopy).

L177,

This calibration is based on a linear trend of absorption coefficients for any type of water ($H_2O$ and OH) in different materials but assumes isotopically distributed molecular $H_2O$ for the orientation factor of 1/3, which is already included in the above equation. The mapped areas also include plagioclase and/or phyllosilicate, whose IR spectra exhibit a dominant band owing to molecular $H_2O$ and accessory band(s) caused by structural OH. It is noted, therefore, that the calibration of Paterson (1982) used for these minerals gives approximate water contents. Other calibrations for contents of molecular $H_2O$ and/or OH species in quartz have been given by Kats (1962), Aines et al. (1984), Nakashima et al. (1995), Libowitzky and Rossman (1997), Stipp et al. (2006), and Thomas et al. (2009). Fukuda and Shimizu (2019) compared these calibrations, and the ratios of calculated water contents between them based on Paterson (1982) as unity are 0.68:0.88:1.15:1.31:1.58:0.43. Previously reported absorption coefficients of water in quartz have been discussed by Fukuda and Shimizu (2019) and Stalder (2021). In this study, water contents are expressed as wt. ppm $H_2O$, which is converted from mol $H_2O$ $\ell^{-1}$ using the molar mass of $H_2O$ (18 g mol$^{-1}$) and the density of quartz (2650 g $\ell^{-1}$). This conversion using these values was applied to other minerals, namely, plagioclase and/or phyllosilicate, which are commonly measured together with quartz in IR maps and used as markers for the locations measured. As the densities (averaged densities when mixed) of these minerals differ from that of quartz, albeit slightly, the water contents of the minerals reported in this study are approximate values.

155-156. This observation, that water content is less between recrystallized and unrecrystallized areas of the same sample is very important. It strengthens conclusions drawn from sample to sample differences in water content. The authors might want to emphasize this more elsewhere in the paper.

As the referee says, this finding is very important. According to the comment, we have added descriptions about "water content is less between recrystallized and unrecrystallized areas of the same sample" as follows.

L21 (Abstract),

These low water contents in recrystallized regions also contrast with those of up to 1540 wt. ppm in adjacent host grains in the weakly deformed sample.

L282 (5.2 Water in dynamically recrystallized quartz in the Discussion section)

These low water contents also contrast with those of up to 1540 wt. ppm in adjacent host grains in the weakly deformed sample.

L399 (Conclusions section),

Water contents in regions of recrystallized grains are 100–510 wt. ppm, with a mean of 220 ± 70 wt. ppm, in the weakly and strongly deformed samples, whereas those of adjacent host grains observed in the weakly deformed sample are up to 1540 wt. ppm.

163-164. It is very difficult, perhaps impossible, to truly distinguish a small subgrain from a small grain in a thick section (when the grain size is much smaller than the thickness of the slide). I don't doubt that there are subgrains, but the language here is misleading.

As the referee says, regions of subgrains are just inferred as we described in the Figure 7 caption. We have revised as follows to describe the existence of subgrains just as a possibility.

L223,

regions of small grains around host grains widen at certain sample stage angles under a polarizing microscope and subgrains would be included (Fig. 8a–f). Water contents in recrystallized regions probably including subgrains are 290–500 wt. ppm (around Nos 1 and 2 in Fig. 8g and h),

169-170. Figure 8a is too zoomed out to see any subgrains (it seems from the placement of "(Fig. 8a)" that we are to see subgrains in the figure).

We have added two enlarged images (one under cross-polarized light and the other under plane-polarized light) that indicate the existence of subgrains. According also to the referee's comment on L163-164, we have changed the sentence to say that subgrains are just inferred.

L231,

The IR mapped area consists mainly of recrystallized quartz (Fig. 9a). Crystallographic orientations of some grains within and around a relic of a host grain look continuously similar and/or continuously change, as in undulose extinction, and these grains would be subgrains (inset under cross-polarized light in Fig. 9a).

We have also revised the Figure 8 caption (Figure 9 in the revised manuscript) as follows.

[Figure]

**Figure 9. IR mapping results…shown with a red square. The two insets in the yellow rectangles show enlarged images of a region that includes a relic of a host grain at the center under cross-polarized (upper) and plane-polarized (lower) light. (b) Water distribution. Numbers … "recrystallized".**

183-184. It is suggested later in the paragraph that small optically invisible inclusions can also provide a similar magnitude IR signal. So why presume that it is the large inclusions that are responsible here? Couldn't it be the case here that the signal is also mainly from submicroscopic inclusions? You could substantiate your claim by measuring the density of inclusions as compared to the IR results in the sample (is there a correlation?), or cite some earlier work if this has been addressed before.

As the referee pointed out, "submicroscopic inclusions" may be included in the almost undeformed sample and they may be measured by IR spectroscopy. In the revised manuscript, we have revised the following sentence; "*a broad band at 2800–3750 cm$^{-1}$ as molecular H$_2$O, which is presumed to correspond to fluid inclusions in quartz grains observed in the optical photomicrograph (Fig. 3b).*" We have focused only on the observations in the optical photomicrographs here.

L246,

The IR spectra commonly show a broad band at 2800–3750 cm$^{-1}$ as molecular H$_2$O in the three different samples. In the almost undeformed sample, fluid inclusions in quartz grains are visible and heterogeneously distributed in the optical photomicrograph (Fig. 4b), but submicroscopic fluid inclusions may be included. In the weakly deformed sample, fluid inclusions in host quartz grains are less numerous in the photomicrograph (Fig. 4d).

185. Change "do not differ" to "do not differ substantially" (means of 500 vs 800 are different)

Revised.

L254,

do not differ substantially

185-187. This is a big claim (that it must be because of invisible inclusions… how would that happen? show us micrograph images of the two areas?). Can the same technique be used to verify the presence of small inclusions that Stunitz used?

The speculation about a redistribution of water by Stünitz et al. (2017) is also based on their observations by polarizing microscope and IR measurements, which are also introduced in our manuscript (L264-L277 in the revised manuscript). This is the same in our study. We suppose that the referee pointed out that invisible or visible inclusions can be included both in the undeformed and weakly samples. On the other hand, the water distributions in the almost undeformed sample are always heterogeneous (Fig. 5; Fig. 6 in the revised manuscript) and they never match with the shapes of quartz grains unlike those in the weakly deformed sample (Figs 7 and 8 in the revised manuscript). Please also see our reply to the referee's comment on L190-193: The referee says in his/her comment on L190-193 that "I imagine that the signal strength might depend on the size of inclusions (microscopic vs sub-microscopic).". If the signal strength depended on the size of inclusions (it does not actually), the signal from invisible inclusions became weaker than that from visible inclusions. Therefore, this part may not make sense for the referee and the referee may have given his/her comment on L185-187.

Regarding "how would that happen?", it is discussed in the papers we cited (Kerrich, 1976; Wilkins and Barkas, 1978; Hollister, 1990; Cordier et al., 1994; Vityk et al., 2000; Faleiros et al., 2010), which are based on observations by polarizing microscope. In the revised manuscript, we have explained the processes proposed by these papers.

L259,

Faleiros et al., 2010): The redistribution of fluid inclusions is possible by volume diffusion and/or diffusion through dislocation cores (pipe diffusion) of water molecules and/or hydrogen.

190-193. I don't know much about IR detection of water, but I imagine that the signal strength might depend on the size of inclusions (microscopic vs sub-microscopic). If so, it would affect this supposition. Add some information on this if possible.

Regarding "I imagine that the signal strength might depend on the size of inclusions (microscopic vs sub-microscopic).", no, it does NOT depend on the size of inclusions but just on the amounts of water measured by the aperture. The calculation of water contents is based on Beer-Lambert law as $A=\varepsilon Cd$, where $A$ is the absorbance, $\varepsilon$ is the absorption coefficient, $C$ is the concentration, $d$ is the sample thickness. Thus, "microscopic vs sub-microscopic" does not matter for "the signal strength", which is converted to the water content. This is basic knowledge about IR spectroscopy (and any types of spectroscopic measurements) so we do not think we need "some information on this".

This comment of the referee and our reply about may be applicable to the referee's comment on L185-187 (and possibly on L183-184 as well). See our reply there.

193-194. Confusing language. Change "are comparable" to "correspond to"

We have revised this expression throughout the manuscript.

L251, L370, L396, L749,

correspond to

195-197. I'm not following the logic here. Why "therefore"? This idea of redistribution seems odd to me. It seems like, overall, water left the system during deformation, so it seems unlikely it would increase in the host crystals—is that what is being implied ("higher water contents…due to the redistribution of fluid inclusions")?

We have deleted the sentence identified by the referee. What we meant was that water in host quartz grains, which are also plastically deformed, is retained and redistributed within host grains, whereas water is released during dynamic recrystallization. We did not mean "it would increase in the host crystals"; as the referee states, "it seems unlikely".

We realized that the original sentence was unclear. Also, according to the referee's comment regarding L183-184, we have reorganized this paragraph in the revised manuscript as follows: 1. Optical micrographs; 2. IR mapping results; 3. Water redistribution based on 1 and 2; and 4. A comparison of our results with previous work on water redistribution.

L247,

In the almost undeformed sample, fluid inclusions in quartz grains are visible and heterogeneously distributed in the optical photomicrograph (Fig. 4b), but submicroscopic fluid inclusions may be included. In the weakly deformed sample, fluid inclusions in host quartz grains are less numerous in the photomicrograph (Fig. 4d). In the IR mapping images for the weakly deformed sample, the high-water-content regions (Figs 7b and 8g) correspond to relics of host grains identified under a polarizing microscope (Figs 7a and 8a–f). In comparison, in the almost undeformed sample, water contents in host grains are simply heterogeneous. As water contents in host quartz grains in the almost undeformed and weakly deformed samples do not differ substantially, although their distributions in the two samples are different, a redistribution of fluid inclusions with a size

change to less than the optical microscopic scale must have occurred in the latter sample. The redistribution of fluid inclusions owing to plastic deformation has been reported in previous microstructural observations (Kerrich, 1976; Wilkins and Barkas, 1978; Hollister, 1990; Cordier et al., 1994; Vityk et al., 2000; Faleiros et al., 2010): The redistribution of fluid inclusions is possible by volume diffusion and/or diffusion through dislocation cores (pipe diffusion) of water molecules and/or hydrogen. Some of those studies proposed leakage of water during the redistribution of fluid inclusions, but this is unlikely to have been significant in the host grains analyzed in the present study since the water contents in host grains in the almost undeformed and weakly deformed samples are not significantly different. The redistribution of fluid inclusions has similarly also been reported from IR spectroscopic measurements of an experimentally deformed single quartz crystal (Stünitz et al., 2017).

210. Section 5.2. It is unclear to me what the main point(s) of this section is. I read it a few times and it feels like a loose collection of thoughts and information from the literature about water in recrystallized quartz. I'm not sure if any new idea is being presented, or what exactly the new data contribute to the previous understanding.
There is limited information on diversity of water contents in regions of recrystallized quartz whose grain sizes are also different. As described in this section, if one compares our water contents and grain boundary widths with those in previous studies, they would know that their values are different. We hope the referee also reads the comment from Referee #3 (Prof. Kronenberg) and understands the importance of this section as well as this study based on IR spectroscopy. Please also see our reply to the referee's comment on Discussion below.

219. "development of dynamic recrystallization" wording is linguistically problematic. Replace with "development of dynamically recrystallized grains" or "dynamic recrystallization".
We have changed to "dynamic recrystallization".
L289,
released by dynamic recrystallization

229. "which may be due to the transition from subgrains to recrystallized grains" This could be tested with EBSD.
We agree but EBSD measurements are beyond of this study. Please also see our reply to the referee's general comment in p. 2-4 of this file.

244. This is a hypothetical statement. Use "were" instead of "is"
We disagree. It does not need to be a subjunctive mood as if it is never realistically possible.

244-245. Just a thought (no need to address it): If the water can be "distributed homogeneously in grain boundaries as thin films," is there any concern that it can be added or lost during the process of making thick sections? What

happens if the samples are heated before IR analysis (this might remove grain boundary water)? Could test for this with IR analysis before and after a heating.

In Fukuda et al. (2009), a paper by the first author, we examined whether water "can be added or lost during the process of making thick sections" and "What happens if the samples are heated before IR analysis (this might remove grain boundary water)". In Fukuda et al. (2009), we used chalcedony, a microcrystalline quartz that consists of grains ranging in size from ~100 nm to 1 μm. The grain size is so small that much more water is stored in grain boundaries compared with grain interiors. We confirmed that water in grain boundaries and pores at the TEM scale is held there during thin sectioning and that tightly trapped water of this type gradually dehydrates when the sample is heated to a temperature of >350 °C (e.g., 20% of water dehydrates during 8 h of heating at 350 °C). Thus, with regard to the referee's questions here, we have already checked that arguments in this manuscript relate to originally trapped water in the samples. We have added the following sentences in Section 3. "Analytical procedure for IR spectroscopy".

L154,

Water in tight grain boundaries and triple junctions (e.g., at the scale of transmission electron microscope resolution), as well as water in grain interiors, is preserved during the above-described processes of thin sectioning and heating on a hot plate (Fukuda et al., 2009). Therefore, arguments and interpretations in this paper concern water that was originally trapped in the samples.

We have added Fukuda et al. (2009) in the reference list.

252-254. This sentence is confusing me. Why is thought to be "continuously supplied"? What is the significance of "textural modifications" here?

This sentence makes a contrast with samples that underwent "textural modifications" ("microstructural modifications" in the revised manuscript) but did not undergo "continuous supply". → (1) This is described in the previous paragraph. The sentence pointed out by the referee here and a few sentences before concern samples that underwent both "textural modifications" and "continuous supply". → (2). We have also described the case of (1) again to clarify, as follows.

L334,

Thus, IR spectra of these samples measured in laboratories may show large water absorption bands stored in grain boundaries with decreasing grain size compared with samples that did not undergo subsequent water infiltration along grain boundaries in previous studies (Finch et al., 2016; Kilian et al., 2016; Kronenberg et al., 2020).

254. "Intracrystaline parts" language is unclear. "Crystal interiors" makes more sense to me (if that is what is meant).

Both "crystal interiors" and "intracrystalline parts" may be linked to the atomic scale of the quartz crystal structure including defect OH; in this case, fluid inclusions may not be included. Therefore, we have changed "Intracrystaline parts" to "grain interiors" throughout the manuscript as Referees #2 and #3 also use this expression in their reviews.
L56, L285, L334, L344, L741, L744,

grain interiors

278. Change "original fluid inclusions can be redistributed" to "water from original fluid inclusions can be redistributed"
Revised.
L361,

water from original fluid inclusions can be redistributed

286-288. I'm not following this logic. Why is diffusion invoked to explain host vs rxld grain water in these samples? All you need is to remove water in recrystallized grains, and the remaining host quartz just holds onto its water (no diffusion necessary except perhaps to redistribute from large to small inclusions).
This sentence discusses diffusion within host grains; it is not about "host vs rxld grain water". To clarify, we have added "within host grains" as follows.
L369,

Thus, the redistribution of fluid inclusions caused by diffusion within host grains may account for water distributions that correspond to the shapes of host grains in the weakly deformed sample (Figs 7 and 8), as discussed in section 5.1.

290-291. The words "that lead to the development of equilibrium texture" may not be necessary, and ideas of "equilibrium" are often complicated (are they really equilibrium? How would we know?). Also it would be necessary to explain what is meant by "equilibrium texture;" do you mean "equilibrium microstructure" (texture for some people means CPO, but I don't think it's what is meant)? Probably best to just trim the sentence by removing these words.
This referee's comment is also associated with his/her comment on L297-299. We realized that we should clearly explain "that lead to the development of equilibrium texture" rather than just trimming "the sentence by removing these words".
We meant "the equilibrium state of water in the system", which is the same in the previous studies we cited for microstructural observations of movements of fluid inclusions by dynamic recrystallization of quartz to form the equilibrium state of water in the system (Kerrich, 1976; Wilkins and Barkas, 1978; Hollister, 1990; Cordier et al., 1994; Vityk et al., 2000; Faleiros et al., 2010). Based on the referee's comments here and on L297-299, we have changed "equilibrium texture" here to "equilibrium state of water in the system" and added explanation about it.
L374,

the equilibrium state of water in the system. In other words, original fluid inclusions in host quartz grains in the photomicrograph (Fig. 4b) and IR spectra (Fig. 6) were trapped during quartz crystallization from magma, and their contents were not equilibrated with the water system under deformation conditions (possibly including water activity and/or water fugacity).

294.-295. I think it is being suggested that formation of subgrains releases an intermediate amount of water. Can this really be inferred with the available data? Consider also the possibility that measuring a mixture of recrystallized grains and undeformed grains also gives an intermediate value. It is very difficult (or impossible) to tell, optically, in a thick (100 μm) sample what is a subgrain and what is a recrystallized grain… EBSD could help clarify what is happening, I believe.

In the original manuscript, we described as follows before the sentence pointed out by the referee here. L219 in the revised manuscript; "In addition, a gradual decrease in water content from the host grains to the adjacent recrystallized regions is observed in some cases; for example, over a distance of ~150 μm from No. 3 (host; 680 wt. ppm in Fig. 7c) to No. 1 (recrystallized region; 210 wt. ppm). In terms of microstructure, this decrease is consistent with the development of dynamic recrystallization from subgrains in and around host grains (Fig. 7a) to recrystallized grains.". And L296; "In addition, in our weakly deformed sample, water contents in quartz gradually decrease with increasing distance from the adjacent host grains within a range of ~150 μm (Fig. 7), which may be due to the transition from subgrains to recrystallized grains. In addition, in the strongly deformed sample, water contents in regions of subgrains are higher than those in recrystallized regions (Fig. 9)."

Thus, regions of subgrains are also inferred even in thick samples and they show higher water contents than those in fully recrystallized regions. Based on the above observations and IR measurements, it seems unrealistic to think about "a mixture of recrystallized grains and undeformed grains" in the IR mapped areas.

297-299. What is the "equilibrium state" of water during deformation? This concept would need to be explored before ending the paper with this sentence. Is it related to temperature, presence of other minerals? Equilibrium between water in quartz and what? Has this been quantified in the literature, and if so what is the significance of the value found (can it be used to quantify anything, water fugacity maybe?)?

As we replied to the referee's comment on L290-291, we have explained "equilibrium state" in L374 in the revised manuscript. Here, we have changed "equilibrium state" to "equilibrium state of water in the system".

L385,

the equilibrium state of water in the system corresponding to the deformation conditions.

629. "addition" not "addiction"

Indeed. Revised.

L748,

**In addition,**

629-633. suggestion: I find the language "former and latter" confusing. Try more direct language, i.e. just say "in the weakly deformed sample" rather than "in the former sample."
Revised.
L748,
the water distributions in host grains in the weakly deformed sample correspond to the shapes of host grains (Figs 7 and 8), but those in the almost undeformed sample do not (Fig. 6).

630. It says water distributions in host grains in the undeformed sample…are "not comparable" with the shapes of host grains (Figs 5–7). This is a strange statement because the size of grains in the undeformed sample is much bigger than the IR maps, so how is this known?
We believe that the expression, "comparable" was confusing. This part was not a comparison between "water distributions" and "the size of grains" but was a comparison between "water distributions" and "the shapes of host grains". According also to the referee's comment on L193-194, we have changed "comparable with" to "correspond to" throughout the manuscript.

Discussion. What is the bigger significance of these findings? It was already known that water can be released during deformation.
We discuss and focus on the following topics. 1. Diversity in water contents with respect to regions of recrystallized quartz and recrystallized grain size. We also state that these water contents and grain sizes are different between our study and previous studies. 2. The release of water during deformation. We emphasize that such release is NOT always common and that a water-infiltration process may have occurred in some samples. 3. The release process of water from subgrains to fully recrystallized grains. Water content in regions of subgrains have not been discussed in previous studies. Please also see our reply to the referee's comment regarding L210.
The referee is familiar with the plastic deformation of quartz and its microstructures. The referee's question here is similar to why scientists continue to observe and measure quartz fabrics, even though we already know about different dynamic recrystallization mechanisms, slip systems, and opening angles of the *c*-axis, among others. Simply speaking, one of the important questions is that we want to know how diverse these aspects are by studying samples that have undergone different crustal conditions. We emphasize that information about water in quartz is in fact very limited, much more so than that about quartz fabrics, for example.

FIgure 1. Label A and B. Need inset showing location in Japan. The lower map should be clear about what the black lines are…are they faults, intrusive contacts, depositional contacts, roads (they seem to be a mix of these

things)? Also the blobs with various deformed status are confusing—is this really the only part of the Wariyama granite that is deformed? Is the deformation related to any of the faults/contacts shown?

In the revised manuscript, we have labeled (a) and (b) and added an "inset showing location in Japan". We have explained "what the black lines are". We have changed one black solid line to a dashed line, which is a forest path including the Takase pass. Other original black lines show geological boundaries. We have explained them in the revised manuscript. The revised figure and caption are shown below.

Regarding "Also the blobs with various deformed status are confusing—is this really the only part of the Wariyama granite that is deformed? Is the deformation related to any of the faults/contacts shown?", they are where the three types of the deformed Wariyama granite are distributed based on our field survey. In the revised manuscript, we have explained the relationship between the samples we studied and Futaba Fault Zone. Please see our reply and revision in p. 7-8 of this file regarding the referee's comment on Section 2 Samples.

[Figure]

The caption for revised Fig. 1 is shown below.

**Figure 1. Study area location map (a) and geological map (b) of northeastern Japan** after Oide and Fujita (1975), Fujita et al. (1988), and Tsuchiya et al. (2014). **The inset in (a) shows the location of the study area and major tectonic lines [the Itoigawa–Shizuoka Tectonic Line (ISTL) and the Median Tectonic Line (MTL)]. In (b), the dashed black line represents a path including Takase pass. Solid black lines represent geological boundaries.** The Wariyama granite was categorized into three types based on the degree of deformation, **as observed in hand specimen** (Fig. 2) and by microstructures observed under a polarizing microscope (Fig**s** 3 **and**4). The three types are referred to as "almost undeformed", "weakly deformed", and "strongly deformed", with distributions as shown in the geological map.

Figure 5, 6, 7, 8. Color scale for the IR maps should be the same so that they can be directly compared visually. For example in line 185-187, it says that regions of two samples have the same water content, but we can't visually "see" this easily the way the figures are currently colored. This would also help compare figures 6 and 7, which are of the same sample. The change will also help the reader intuitively grasp the main result of the paper (the strongly deformed sample will show up as mostly blue, whereas the undeformed sample will be red).

We originally considered presenting all of the IR maps with the same color scale, as suggested by the referee. The same color scales were also used, for example, by Kronenberg et al. (2017). As the referee states "(the strongly deformed sample will show up as mostly blue, whereas the undeformed sample will be red)", which is true. However, in our study, this color-coding option makes detailed water distributions obscure, especially for Figs 6 and 8 (Figs 7 and 9 in the revised manuscript). Please see the figures presented below, which show original figures and alternative figures in the same color scales.

In the alternative figures of the IR maps, we used the same color scales based on Fig. 5c (Fig. 6c in the revised manuscript), in which the red color for the maximum water content represents 1800 wt. ppm (other alternative Figs 6-8c show the red color as >1800 wt. ppm because of the much higher water contents in plagioclase).

In alternative Fig. 6c, the difference between the measured point No. 1 and 2 cannot be visually distinguished, nor can the water distribution around No. 3 (regions of subgrains). Therefore, the following explanation in the original manuscript would have been unclear. L218 in the revised manuscript; "In addition, a gradual decrease in water content from the host grains to the adjacent recrystallized regions is observed in some cases; for example, over a distance of ~150 μm from No. 3 (host; 680 wt. ppm in Fig. 7c) to No. 1 (recrystallized region; 210 wt. ppm)."

Alternative Fig. 7c can be directly compared with Fig. 6c, so the alternative color coding may be better for this figure. In alternative Fig. 8c, water contents around the measured points Nos 3 and 4 (regions of subgrains) are not clear, which is problematic.

In addition, if the same color scales were applied to all of the IR maps, we would be concerned that readers may try to equate reddish parts with high water contents in quartz, which does not exist in most of the IR maps.

Thus, because of the above reasons regarding the same color scale (that is, the lack of clarity about water distributions that we discuss and possible confusion for readers), we adopted different color scales for each IR map.

[Figure]

Figure 5 (Figure 6 in the revised manuscript).

[Figure]

Figure 6 (Figure 7 in the revised manuscript).

[Figure]

Alternative Figure 6c.

[Figure]

Figure 7 (Figure 8 in the revised manuscript).

[Figure]

Alternative Figure 7c.

[Figure]

Figure 8 (Figure 9 in the revised manuscript).

[Figure]

Alternative Figure 8c.

**Some questions I am left with:**

How significant would the change in water content be in terms of rheology of the samples?

The undeformed quartz is heterogeneous in terms of water content. Water weakens quartz. So one might expect that deformation would occur mainly in areas of high water content. However, the host quartz remaining in weakly deformed quartz has more water, on average, than the undeformed quartz—not what you'd expect if recrystallization were focused in the high-water areas. How do you explain this? Perhaps it is a result of the small data set (i.e. one of the samples is anomalous in terms of water content)

It is difficult not only from this study but also from other similar studies to discuss the relationship between water contents and rheology. As we discussed in Section 5.2, we compared our results with those in Finch et al. (2016), Kilian et al. (2016), and Kronenberg et al. (2020) for naturally deformed quartz and Palazzin et al. (2018) for experimentally deformed quartz. Finch et al. (2016) has given a possibility of transition from dislocation creep to diffusion creep due to released water but other studies did not see an effect(s) of released water. The referee says this may be because of "a result of the small data set", which would not be applicable only to our study but also to all similar studies. As we replied to the referee's comments on L210 and Discussion, there is still only limited information on water contents, species, and distributions with the development of texture in deformed quartz.

*Replies to Referee #2's comments*

Comments on the manuscript: "Water release and homogenization by dynamic recrystallization of quartz" by Fukuda, Okudaira, and Ohtomo

The manuscript describes FTIR measurements in quartz from granitoids deformed to different finite strains. The authors find that the $H_2O$ content decreases systematically with different degrees of dynamic recrystallization due to different finite strains. The manuscript is concisely written, to the point, and the conclusions are supported by the data presented. I recommend that the manuscript is published after minor revisions.

We thank positive comments. According to the referee's comments shown below, we have revised the manuscript.

Detailed comments:

Line 34: ad reference: Negre et al. 2021

Done. We have also added it in the reference list.

L35,

Lusk et al., 2021; Nègre et al., 2021).

Line 50: please add "quartz aggregates" instead of "in quartz" to include grain boundaries rather than just grain interiors.

Revised.

L52,

IR spectroscopic measurements of water in quartz aggregates have revealed that

Line 53: better "grain boundary regions" than just grain boundaries, because you consider volumes here (grain boundaries would just be surfaces).

Revised.

L55,

grain boundary regions

Line 78: instead of using "texture", it would be better to use "microstructure" or "fabric" here. Texture in a deformation context could be confused with CPO. This applies to the whole text.

We have changed "texture(s)" or "textural" to "microstructure(s)" or "microstructural" throughout the manuscript.

Line 155: better: "...host quartz grains..."

Revised.

L215,

host quartz grains

Line 156: some recrystallized regions contain H2O higher than 300 ppm H2O (light blue colors correspond to about 400 ppm H2O in Fig. 6).

Water contents "higher than 300 ppm H2O" develop "within ~150 μm around host grains": These regions may be composed of subgrains and were described next. We have clarified these facts as follows.

L216,

In contrast, water contents in the recrystallized regions of >~150 μm away from the adjacent host grains are 200–300 wt. ppm, clearly lower than those in the host grains. In addition, a gradual decrease in water content from the host grains to the adjacent recrystallized regions is observed in some cases; for example, over a distance of ~150 μm from No. 3 (host; 680 wt. ppm in Fig. 7c) to No. 1 (recrystallized region; 210 wt. ppm). In terms of microstructure, this decrease is consistent with the development of dynamic recrystallization from subgrains in and around host grains (Fig. 7a) to recrystallized grains.

Lines 162-166: the "subgrains" could be regions with healed microscracks – it is difficult to see from Fig. 7b. Higher H2O contents in such regions would be consistent with microcracks. Please mention the possibility of microcracks and perhaps discuss them later in the discussion section.

We have added enlarged images for the region including recrystallized grains and possible subgrains. We do not observe healed microcracks even in the enlarge images but the existence of microcracks is not completely denied. Therefore, we have revised as follows.

[Figure]

Revised Figure 7 (Figure 8 in the revised manuscript).

L223,

In another part of the weakly deformed sample (Fig. 8), regions of small grains around host grains widen at certain sample stage angles under a polarizing microscope and subgrains would be included (Fig. 8a–f). Water contents in recrystallized regions probably including subgrains are 290–500 wt. ppm (around Nos 1 and 2 in Fig. 8g and h), slightly higher than those in the recrystallized regions in Fig. 7. These regions look slightly darker under plane-polarizing light, probably because of

light scattering by fine grains (Fig. 8c and e). We did not observe microcracks that hold water, at least under a polarizing microscope.

Caption,

**Figure 8. IR mapping results for the weakly deformed sample. The sample thickness is 97 μm. (a) Optical photomicrograph under cross-polarized light. The IR mapped area is shown with a red square. The dashed white rectangle depicts the area of enlarged images of fine-grained quartz regions including host grains under (b) cross-polarized and (c) plane-polarized light. Similarly, images (d) and (e) are from (f), where the optical photomicrograph is rotated approximately 30° clockwise from (a).**

Line 220: add reference Kilian et al. 2016.

Done.

L289,

Finch et al. (2016), Kilian et al. (2016), and Kronenberg et al. (2020)

We have also introduced the water contents and species in quartz reported in Kilian et al. (2016). Their water contents were significantly lower than those in our study and other previous studies (Finch et al. 2016; Kronenberg et al. 2020). We have added the following sentences.

L307,

Kilian et al. (2016) studied quartz in a granite deformed under amphibolite-facies conditions. In their samples, quartz grains were dynamically recrystallized to a grain size of 250–750 μm mainly by grain boundary migration with a minor contribution by subgrain rotation. Their mean water contents, including intracrystalline OH and molecular $H_2O$ as fluid inclusions in recrystallized quartz, were ~10 wt. ppm, comprising 70%–80% of molecular $H_2O$ and the remainder of intracrystalline OH, and were approximately half of those in the original magmatic host grains. Those values in recrystallized and original grains are much lower than those in other previous studies and in the present study. Thus, water contents in recrystallized quartz grains and host grains are diverse in natural samples; this diversity may depend on deformation conditions, including the equilibrium state of water.

**Replies to Referee #3's comments (Prof. A. Kronenberg)**

This manuscript presents IR absorption spectra of OH stretching bands and maps of water content measured for deformed granite samples from the Wariyama uplift of NE Japan, including a Wariyama granite that is not deformed or only modestly deformed (designated as "almost deformed"), and weakly deformed and strongly deformed granites of the same unit. The deformation microstructures of quartz in these samples is described, including undulatory extinction, subgrains and recrystallized grains with characteristics of dynamic recrystallization during dislocation creep. I am pleased to see that the authors have found similar results for water in quartz during recrystallization as our group did for deformed and recrystallized quartz mylonites (Kronenberg et al., 2020). I will comment, in part, in response to remarks and discussion of other review that suggest that the results here are not new. I would counter that it is necessary to learn whether recrystallization routinely excludes water from quartz grain interiors or not. The fact that Finch et al (2016) found reductions in water content with strain in the El Pichao shear zone of NW Argentina, and we found reductions in water contents with recrystallization of the Moine Thrust, NW Scotland is no guarantee that water contents will be reduced in other deformed rocks undergoing dislocation creep and recrystallization. In other words, I'm pleased to learn that this result was obtained in the present study and with further confirmations, we may be able to generalize the conclusion that water contents always (or normally) decrease with recrystallization.

We thank positive comments and encouragement. Indeed, there is very limited information on behavior of water in quartz deformed by plastic deformation. We are pleased to hear that the referee agrees with the importance of our study.

On the other hand, I find that this manuscript includes new results for coexisting plagioclase grains of the same deformed granites, and I recommend that the authors revise the manuscript for publication, building on these results. Firstly, neither the title nor abstract prepare the reader for the interesting IR results for plagioclase of these deformed granites. The results section includes IR spectra for plagioclase but the discussion section could address these results further. Secondly, I cannot tell from the manuscript whether feldspars of these granites are internally deformed or not, and whether they are also recrystallized. The IR spectra show that coexisting plagioclase grains are very wet, compared with deformed quartz grains, and their spectra appear to include sharp high-wavenumber OH bands due to layer silicate (I assume sericite) inclusions. In an earlier IR study of Sierra Nevada granites deformed at greenschist grade conditions, Kronenberg et al. (1990) also found that feldspars had higher water contents than quartz grains of the same granites. However, we found no evidence of plastic deformation of the feldspars so we could not say anything about water weakening of feldspars. Apparently the temperature of deformation was too low for any plastic deformation of feldspars.

I propose the following additions to this contribution: 1) describe the microstructures of plagioclase relevant to their brittle or plastic deformation, and any recrystallization that might have occurred (if no plastic deformation is evident, that's OK – simply report your observations), 2) if the plagioclase is recrystallized at grain margins, are they wet in these regions or are they dry? 3) please cite any papers or results that provide constraints on metamorphic temperatures during Wariyama granite deformation. Again, providing a link between temperature/metamorphic facies, deformation and water contents of feldspars and comparisons with the dislocation glide and creep of quartz would be very useful, and 4) please describe the fluid inclusions and white mica inclusions in plagioclase grains that relate to their large OH absorption bands. I would be curious if these rocks also reveal evidence of plastic deformation and recrystallization of plagioclase and potential water weakening of feldspars at conditions that favor dislocation creep, subgrain formation, and dynamic recrystallization of quartz. 5) With such results, the authors can also compare with results reported for other deformed granitic rocks. For example, Kilian et al. (2016) found that quartz deformed and recrystallized at higher temperature, amphibolite facies conditions are very dry. Feldspars in those same rocks were observed to be less deformed (appearing as augen) but they were recrystallized at their margins. The previous study did not characterize OH of feldspars but this study does, so adding the microstructural information is important.

As the referee states in comments 1), 2), and 3), we realized that we did not mention the deformation of plagioclase and deformation conditions in the original manuscript, even though we showed some IR spectra of plagioclase. With regard to the referee's comments 1) and 2), brittle deformation of plagioclase is dominant and no recrystallization is observed. We have added a new figure (Fig. 3 in the revised manuscript) that shows the general microstructures of the samples and have provided descriptions of the deformation of plagioclase. With regard to the referee's comment 3), we have also discussed deformation conditions based on this new figure in the revised manuscript.

L86,

In the almost undeformed and weakly deformed samples, mafic igneous minerals are partly replaced by epidote and chlorite (Fig. 3a–d). Although some or all chlorite may have replaced amphibole during late-stage hydrothermal alteration in the three types of sample (Fig. 3a, c, and e), chlorite is found in shear bands in the strongly deformed sample (Fig. 3e), suggesting that chlorite may have been stable during mylonitization. Grain interiors of plagioclase are commonly altered to epidote, muscovite, and clay minerals in the three types of sample. In the weakly and strongly deformed samples (Fig. 3c–f), brittle deformation of plagioclase dominates and plagioclase and amphibole form porphyroclasts, but there is no development of microstructures indicative of dynamic recrystallization or pressure shadows at their margins. In the strongly deformed samples, amphibole (mainly hornblende) and epidote grains may have been stable during mylonitization because shear bands developed in the samples are composed of amphibole and epidote grains (Fig. 3e and f). According to Fujita et al. (1988), some metasedimentary rocks affected by the thermal effects of granitoid intrusions in the Wariyama area contain metamorphic andalusite, implying

that these granitoids were emplaced into upper-crustal levels. On the basis of these observations, mylonitization at least of the strongly deformed sample is inferred to have occurred under epidote–amphibolite-facies conditions and within or near the andalusite stability field (i.e., ~500 °C; Spear, 1993).

L120,
Differences in the recrystallization mechanisms of quartz in the three types of sample may be due to differences in temperature, strain rate, and/or stress.

[Figure]

**Figure 3. Optical photomicrographs showing microstructures of the Wariyama granite under plane-polarized (a, c, and e) and cross-polarized (b, d, and f) light. (a and b) Almost undeformed sample. (c and d) Weakly deformed sample. (e and f) Strongly deformed sample. Microstructures of quartz in the three samples are shown in Fig. 4.**

With regard to the referee's comments 3) and 4), and as suggested by the referee, we have generated two example figures below from Figs 6 (Fig. 7 in the revised manuscript) and 7 (Fig. 8 in the revised manuscript) showing variations in integral absorbance at 2800–3750 cm$^{-1}$ for the broad band of water and at 3550–3700 cm$^{-1}$ for the OH peak seen in IR spectra. Plagioclase grains are largely altered to epidote, muscovite, and clay minerals, as we described above together with new Fig. 3 in the revised manuscript. It is likely that heterogeneities in the IR maps below may be due to amounts of clay minerals altered from plagioclase as well as the amounts of quartz measured together. Thus, IR spectra of plagioclase may not give useful information that is comparable to behavior for the plastic deformation of quartz or the brittle deformation of plagioclase. We consider that an explanation of this fact should be sufficient, rather than showing these IR maps in the revised manuscript. In addition to the explanation about brittle deformation and alteration of plagioclase for the new Fig. 3, we have also given the following explanation about IR spectra of plagioclase.

L211,

In all of the studied samples, grain interiors of plagioclase are altered to epidote, muscovite, and clay minerals (Fig. 3). When ductile shear zones developed in the study area, plagioclase was not deformed plastically but was replaced by various minerals during late-stage alteration. Therefore, the IR spectra of plagioclase that includes altered minerals may not provide useful information about water in rocks during mylonitization.

[Figure]

**Figure caption: IR mapping results for the mapped area in Fig. 7 in the revised manuscript for integral absorbance at 2800–3750 cm⁻¹ (a, total water region) and 3550–3700 cm⁻¹ (b, sharp OH band region). Numbers in (b) represent raw IR spectra shown in (c). Linear baselines in these wavenumber regions are depicted by dashed red lines. Minerals included in the measurements were judged from the optical photomicrograph and structural vibrations at <2000 cm⁻¹ and are shown with values of integral absorbance in the two wavenumber regions.**

[Figure]

**IR mapping results for the mapped area in Fig. 8 in the revised manuscript for integral absorbance at 2800–3750 cm⁻¹ (a, total water region) and 3550–3700 cm⁻¹ (b, sharp OH band region). Numbers in (b) represent raw IR spectra shown in (c). Linear baselines in these wavenumber regions are depicted by dashed red lines. Minerals included in the measurements were judged from the optical photomicrograph and structural vibrations at <2000 cm⁻¹ and are shown with values of integral absorbance in the two wavenumber regions.**

In the following, I comment and ask the authors to correct or clarify technical issues.

First, the ability to measure IR spectra using very small apertures (25 x 25 microns) using a conventional FTIR and IR source (not IR of a synchrotron) is impressive and this leads to nice OH maps of the samples.

It would be useful to the reader to describe the use and characteristics of the glycol phthalate resin used in preparation of thin sections. I assume this resin was used to inject samples in order to polish the IR samples. However, I do not know anything about the IR spectra of this substance or whether it can be completely removed from samples after polishing is complete. Please describe the use of this resin and whether it represents an issue for the spectra of quartz and plagioclase.

According to the comment, we have carefully described the use and characteristics of the resin. Because the resin as well as acetone is an organic compound, CH peaks around 2950 cm$^{-1}$ can be an indicative of their contamination. We did not observe them so these substances are completely removed or at least below the detection limit in IR spectra.

L140,

The process for making thin sections followed that for regular thin sections with a thickness of ~30 μm for observation using a polarizing microscope but there are a few exceptions: One side of the sample was polished down to #6000 aluminum oxide powder and glycol phthalate resin was used to attach the polished sample surface on a glass slide on a hot plate heated to 100–120 °C. Then, the other side of the sample section was polished down to #6000 to give an approximate thickness of 100 μm to minimize scattered IR light on the sample surface and interference fringes within the sample. Finally, the sample was removed from the glass slide by dissolving the resin in acetone. IR spectra obtained in this study did not show any CH peaks around 2950 cm$^{-1}$, which can be indicative of organic contamination from residuals of the resin and/or acetone; even if these peaks are detected in IR spectra, they do not affect water bands, as the wavenumbers are different (e.g., Kebukawa et al., 2009).

We have added Kebukawa et al. (2009) in the reference list.

Also, we measured IR spectra of the resin pasted on a glass slide. The results are shown below. We assumed undissolved amount of 1% as an example, which corresponds to the resin thicknesses of 1 μm for the actual sample thickness of ~100 μm (left) and 100 μm for the sample thickness normalized to 1 cm in the manuscript (right). In the left figure, an intense peak around 1950 cm$^{-1}$ due to CH vibrations is observed. However, as in the right figure, the whole absorbance is far lower that of water in quartz. The actual raw IR spectra shown in the manuscript (Figs 6-9) do not show CH peaks. Thus, the resin is completely removed or at least below the detection limit in IR spectra.

[Figure]

The authors state very clearly their choice of the Paterson relationship between IR absorbance and OH contents for quartz grains, and comparisons between this relationship and other calibrations for quartz. However, the authors should state clearly that this calibration depends on integrated absorbance, and shape of the OH absorptions. Was the Paterson relationship and its method of integrating also used for the plagioclase grains?

According to the comment, we have stated that "this calibration depends on integrated absorbance, and shape of the OH absorptions.". To express water contents as wt. ppm in quartz, we used the molar mass of $H_2O$ (18 g mol$^{-1}$) and density of quartz (2650 g $\ell^{-1}$), which were applied to all of the mapped data. Therefore, water contents in plagioclase as well as phyllosilicate included are rough values but these minerals were used as markers for the locations mapped by IR spectroscopy.

L177,

This calibration is based on a linear trend of absorption coefficients for any type of water ($H_2O$ and OH) in different materials but assumes isotopically distributed molecular $H_2O$ for the orientation factor of 1/3, which is already included in the above equation. The mapped areas also include plagioclase and/or phyllosilicate, whose IR spectra exhibit a dominant band owing to molecular $H_2O$ and accessory band(s) caused by structural OH. It is noted, therefore, that the calibration of Paterson (1982) used for these minerals gives approximate water contents. Other calibrations for contents of molecular $H_2O$ and/or OH species

in quartz have been given by Kats (1962), Aines et al. (1984), Nakashima et al. (1995), Libowitzky and Rossman (1997), Stipp et al. (2006), and Thomas et al. (2009). Fukuda and Shimizu (2019) compared these calibrations, and the ratios of calculated water contents between them based on Paterson (1982) as unity are 0.68:0.88:1.15:1.31:1.58:0.43. Previously reported absorption coefficients of water in quartz have been discussed by Fukuda and Shimizu (2019) and Stalder (2021). In this study, water contents are expressed as wt. ppm $H_2O$, which is converted from mol $H_2O$ $\ell^{-1}$ using the molar mass of $H_2O$ (18 g mol$^{-1}$) and the density of quartz (2650 g $\ell^{-1}$). This conversion using these values was applied to other minerals, namely, plagioclase and/or phyllosilicate, which are commonly measured together with quartz in IR maps and used as markers for the locations measured. As the densities (averaged densities when mixed) of these minerals differ from that of quartz, albeit slightly, the water contents of the minerals reported in this study are approximate values.

It has become common practice to report water contents of minerals as weight ppm H2O rather than molar (or atomic) ppm (H/Si) but some OH absorption bands of quartz are due to hydrogen interstitials (OH) and do not represent H2O defects. In addition, the original measurements by which OH absorption bands of standards have been determined (such as calibrations of Kats 1962) are not necessarily measurements of H2O weight per oxide (SiO2) weight. As I result, I don't favor this choice of concentration units of measure, even though the authors have every right to use the units of choice that are now in wide usage. More importantly, the authors should clarify how they map water contents in their maps when some of the map area consists of quartz and some consists of plagioclase. Have separate calibrations been used for quartz and plagioclase to map water contents? The contouring is in wt ppm, but I find it difficult to compare these for two minerals with different formula weights. This becomes even more complicated for plagioclase grains that include a sharp band at 3630 cm-1 due to sericite inclusions, which represent OH of layer silicates and are probably strongly polarized relative to the crystallographic axes of the sericite inclusions. The methods of determining and plotting H2O over areas of multiple phases should be described. I wonder if it isn't safer to map and contour integrated absorbances over OH bands of the mapped area, and infer water contents from these maps, rather than attempt to plot water contents using multiple conversions of OH absorption to water content.

We know that some people prefer the ppm H/Si unit because of the reasons that the referee commented. When only "hydrogen interstitials (OH)" is discussed as in Kats (1962), the use of ppm H/Si can be valid. On the other hand, this study and some studies after Kats (1962) deal with molecular $H_2O$ as fluid inclusions and in grain boundaries. In this case, the ppm H/Si unit, which may infer "hydrogen interstitials (OH)", seems odd for us although there may be some association between molecular $H_2O$ and hydrogen interstitials as in their mutual transformations (Stünitz et al., 2017) but may not always. Therefore, we used the wt. ppm unit in our study.

Regarding "Have separate calibrations been used for quartz and plagioclase to map water contents?", we did not separate calibrations for quartz and plagioclase; in the aperture size of 25 x 25 μm, plagioclase as well as phyllosilicate is often measured together with quartz. We could have performed IR mapping measurements or just

shown the IR results only for quartz but we were worried that readers might not be able to visually see where the mapped areas were by comparing the IR mapped data with the photomicrographs. Then, minerals other than quartz were mainly used markers for the mapped locations. Another (minor) reason was that readers might be interested in IR spectra of other minerals so we showed their spectra as well. To determine water contents in wt. ppm, densities of minerals are needed and we have also clarified the method to determine wt. ppm in the revised manuscript. Again, quartz, plagioclase, and phyllosilicates are often measured together so their accurate densities are unknown. However, the density of quartz is 2650 g $\ell^{-1}$ and those of plagioclase and phyllosilicates are around 2700 g $\ell^{-1}$ and 3000 g $\ell^{-1}$, respectively. Since these values are not significantly different, water contents for the latter two minerals and mixtures of them with quartz using the quartz density can give rough values. We have also clarified it in the revised manuscript.

As the referee says, we could only show "integrated absorbances over OH bands of the mapped area, and infer water contents from these maps" as we showed above. However, because readers can directly see water contents in quartz in the IR mapped data, we prefer to keep the original IR maps in wt.ppm.

To sum up, in the revised manuscript, we have given careful descriptions of how we determined water contents (in wt. ppm) and why these estimations can give approximate values for water contents in plagioclase and phyllosilicates, which are often measured with quartz. The revised parts are shown above regarding the referee's comment on the use of Paterson's (1982) calibration (L177 in the revised manuscript).

There are a few places that the author might reword the manuscript text.

For example, in the Samples section, I recommend replacing "by the naked eye" by "as observed in hand specimen"
According to the comment, we have revised the expression in the manuscript.
L84,

on the basis of the development of foliations of mylonitic rocks as observed in hand specimen

L684,

three types based on the degree of deformation, as observed in hand specimen

In the same section, I recommend replacing "as an equivalent circle diameter" by "as the diameter of a circle of equivalent area to the diameter of the grain"
Revised.
L125,

as the diameter of a circle of equivalent area to the diameter of the grain

In the section on Analytical Procedure for IR spectroscopy, I assume that the sample translation stage translates the sample in X-Y while the IR beam was fixed in place (not as described by a "beam-moving function").

The IR beam is actually moved by adjusting mirror positions in the apparatus and "the sample translation stage" is fixed. We have clarified as follows.

L165,

Mapping measurements were performed using a beam-moving function in the sample area of up to 400 × 400 μm; the sample stage was fixed, and IR light irradiated to the sample was moved.

In section 5.1 of the Discussion, I am not sure what is meant by "OH dislocations". Do you mean OH at dislocation cores? Or do you mean a fully hydrated dislocation (OH completely saturate dangling bonds of dislocations)?

We meant Si-OH in the quartz crystal structure and it is not necessarily fully hydrated. We have clarified it based on the discussion in Stünitz et al., (2017).

L264,

The redistribution of fluid inclusions has similarly also been reported from IR spectroscopic measurements of an experimentally deformed single quartz crystal (Stünitz et al., 2017). Those authors demonstrated that original fluid inclusions with sizes of up to 100 μm transformed into Si–OH as Si–O–Si + $H_2O$ ← → Si–OH ⋯ OH–Si, where Si–OH showed a sharp band at 3585 cm$^{-1}$ in IR spectra. … Stünitz et al. (2017) also demonstrated that the above Si–OH, which could have originally been visible fluid inclusions with sizes of a few micrometers under an optical microscope, are again transformed into much smaller fluid inclusions measuring less than 100 nm in size by subsequent annealing.

Referring to the last line of the Section 5.3 of the Discussion, I agree with the authors and doubt that many water contents measured in deformed quartz actually represent equilibrium concentrations. They are generally too large for equilibrium defects and highly variable spatially.

Thank you for supporting our idea. Water especially in undeformed quartz, where water diffusion can be slow, may be trapped during the crystallization stage from magma and may not reflect the equilibrium concentrations during the deformation stage.

Caption to Figure 9 – minor typo- "addiction" should be respelled as "addition"

Revised.

L747,

**In addition,**

Commenting on Data Availability, are the Jasco data files formatted in a way that readers can download Jasco-formatted IR files and plot the results as open-source files without purchasing special, analysis software? If so, great. If not, it would be good to make files available as csv or other open-source file formats.

The Jasco original data files with extensions of ".jwa" require special software to open. Therefore, in the data sets, we also uploaded text files for those who do not have the software. In the revised manuscript, we have clarified this point.

L410,

The text files can be used for those who do not have JASCO's IR software.

In summary, this is an interesting contribution that will help us understand deformation of quartz and feldspars at natural strain rates. The results for quartz appear to support and corroborate previous reports of water losses from shear zones undergoing dislocation creep and recrystallization, and the results for plagioclase are new. The plagioclase results will have increased impact on the field by including the same detail of microstructural descriptions for plagioclase as for quartz.

The scientific quality of the results presented is very good, but the methods, as discussed above, need to be more clearly stated, and water contents and microstructures of plagioclase need further evaluation and representation, to the same extent as presented for quartz. The changes I propose are not truly major but they involve more than a simple rewrite, so I characterize them as "major".

We have added a new figure as Fig. 3 to discuss microstructures of plagioclase and other minerals that can show the deformation behavior and conditions of the samples studied in this study. As we described above, brittle deformation of plagioclase is dominant and grain interiors of plagioclase are altered to clay minerals. In this response file, we showed IR maps and IR spectra focusing grain interiors of plagioclase in terms of integral absorbance at 2800-3750 cm$^{-1}$ for the broad band of water and at 3550-3700 cm$^{-1}$ for the OH peak around 3630 cm$^{-1}$. As a result, as we described above, these IR data for plagioclase do not give useful information that can be compared with or applied to plastic deformation of quartz which is focused on in this study. Therefore, we would like to keep them in this response file. The referee was interested in water in plagioclase and we realized that we had not carefully described deformation of plagioclase as well as other minerals in the studied samples in the original manuscript. Therefore, we have carefully described information about plagioclase, other minerals, and deformation conditions in the revised manuscript.

I look forward to seeing this paper in print,

Andreas Kronenberg

---

## Author Response (AR2)

Referee's comments: Black (Arial)

Our replies: Blue (Arial)

Revisions made in the revised manuscript-1: Red (Times New Roman).

Revisions made in the revised manuscript-2 (this time): Green (Times New Roman).

Descriptions from the original manuscript: Black (Times New Roman)

There were comments only from Referee #3 this time. His comments, our replies, and revisions in the manuscript are shown below.

**Replies to Referee #3's comments (Prof. A. Kronenberg)**

This manuscript shows that OH contents are reduced during recrystallization, corroborating similar results in other shear zones, and helps build the case that this is a phenomenon that is widespread.

The authors have responded to my earlier concern that OH contents of plagioclase and other minerals seem to have been reported using the same calibration as for quartz of the deformed rocks. While it is true that the Paterson calibration is based on a wide range of hydrous materials, there is still a problem reporting OH contents as ppm by weight. What weight is assumed for intermediate plagioclase? The authors note that further analysis is not warranted of the plagioclase grains due to alteration of these grains. This brings up further issues, when OH spectra include OH bands of epidote and layer silicates. Layer silicates, for example, have extremely strong and polarized OH bands. Thus, the assumption of isotropic OH bands with the Paterson relationship cannot give good results when OH bands are not measured for specific crystalline optical vibrational directions.

The authors use the word "approximate" to describe the OH contents of phases other than quartz. I suggest strengthening this a bit - you can detect variations in OH spatially withint plagioclase or other grains, but the OH contents are qualitative, not approximate. This shouldn't be such a bad thing, though, as the focus of the contribution is clearly on OH contents of quartz. And in this regard, the Paterson calibration (assuming isotropic OH bands) is reasonable given that the broad OH absorption of quartz that correlates well with mechanical properties is isotropic.

With regard to the referee's comment in the previous paragraph, indeed, structural OH of epidote, layer silicates, and plagioclase would show polarized OH bands. This means that structural OH is incorporated anisotropically in these minerals. We have given this explanation in the revised manuscript. Then, as the referee says, we realized that the word "approximate" is inappropriate. On the other hand, the word "qualitative" may not discuss "contents" but "compositions", which in our case can be "OH band(s), $H_2O$ band, their wavenumber positions, and/or their ratios, etc.". We consider that "semi-quantitative" would best fit with our case because the calibration of Paterson

(1982) gives some water content values for the above minerals which however may not be accurate. We have revised the relevant text as follows.

L177,

This calibration is based on a linear trend of absorption coefficients for any type of water ($H_2O$ and OH) in different materials but assumes isotropically distributed molecular $H_2O$ for the orientation factor of 1/3, which is already included in the above equation. The mapped areas also include plagioclase and/or phyllosilicate. Grain interiors of plagioclase are also altered to epidote, muscovite, and clay minerals (Fig. 3 and see the description in Section 2). IR spectra of plagioclase possibly including these alteration minerals as well as those of phyllosilicate exhibit a dominant band owing to molecular $H_2O$ and accessory band(s) caused by structural OH (shown later). Especially, structural OH may be anisotropically incorporated in all of these minerals. It is noted, therefore, that the calibration of Paterson (1982) used for these minerals gives semi-quantitative water contents.

L198,

the water contents of the minerals reported in this study are semi-quantitative values.

One minor mis-spelling that needs correction: on line 188 of the newly revised manuscript, "isotropic" is mispelled as "isotopic", which a spell checker won't find, but it not the intended word.

Revised.

L179,

assumes isotropically distributed molecular $H_2O$

In addition to the above revisions according to the referee's comments, we have revised the following text.

L217,

alteration minerals

We have changed "altered" to "alteration", which is more commonly used.

In the Data availability section, we will officially publish our data in Mendeley Data, and delete the following italicized part at the time of proofreading.

L409,

Raw IR mapped data for Figs 6–9 in JASCO format (e.g., Fig6.jwa) and each exported IR spectrum for each sample position (e.g., Fig6_X1Y1.txt) are available from Mendeley Data, V1, doi: 10.17632/zn24kbg9xt.1 *at https://data.mendeley.com/datasets/zn24kbg9xt/draft?a=3552fbc3-364c-4edd-b50a-fa58d198bcce [Comment from authors:*

*This is a temporary shared URL. We will officially publish the data in Mendeley Data when the manuscript is published as an article in Solid Earth].*

We have revised the Figure 1 caption as follows; "the location of the study area" → "the location of (a) in Japan".
L684,
The inset in (a) shows the location of (a) in Japan and major tectonic lines…